# Human brain state dynamics are highly reproducible and associated with neural and behavioral features

**Kangjoo Lee** [1]*, **Jie Lisa Ji** [1], **Clara Fonteneau** [1], **Lucie Berkovitch** [1,2,3,4], **Masih Rahmati** [1], **Lining Pan** [1], **Grega Repovš** [5], **John H. Krystal** [1], **John D. Murray** [1,6,7,8], **Alan Anticevic** [1,6,9]

1 Department of Psychiatry, Yale University School of Medicine, New Haven, Connecticut, United States of America, 2 Saclay CEA Centre, Neurospin, Gif-Sur-Yvette Cedex, France, 3 Department of Psychiatry, GHU Paris Psychiatrie et Neurosciences, Service Hospitalo-Universitaire, Paris, France, 4 Université Paris Cité, Paris, France, 5 Department of Psychology, University of Ljubljana, Ljubljana, Slovenia, 6 Interdepartmental Neuroscience Program, Yale University School of Medicine, New Haven, Connecticut, United States of America, 7 Department of Physics, Yale University, New Haven, Connecticut, United States of America, 8 Department of Psychological and Brain Sciences, Dartmouth College, Hanover, New Hampshire, United States of America, 9 Department of Psychology, Yale University, New Haven, Connecticut, United States of America

* kangjoo.lee@yale.edu

## Abstract

Neural activity and behavior vary within an individual (states) and between individuals (traits). However, the mapping of state-trait neural variation to behavior is not well understood. To address this gap, we quantify moment-to-moment changes in brain-wide co-activation patterns derived from resting-state functional magnetic resonance imaging. In healthy young adults, we identify reproducible spatiotemporal features of co-activation patterns at the single-subject level. We demonstrate that a joint analysis of state-trait neural variations and feature reduction reveal general motifs of individual differences, encompassing state-specific and general neural features that exhibit day-to-day variability. The principal neural variations co-vary with the principal variations of behavioral phenotypes, highlighting cognitive function, emotion regulation, alcohol and substance use. Person-specific probability of occupying a particular co-activation pattern is reproducible and associated with neural and behavioral features. This combined analysis of state-trait variations holds promise for developing reproducible neuroimaging markers of individual life functional outcome.

## Introduction

The field of functional Magnetic Resonance Imaging (fMRI) has attempted to characterize the functional organization of the human brain and how it relates to individual differences [1,2]. These emerging methods can identify low-dimensional representations of neural traits (i.e., subject-specific) [3,4] or states (i.e., varying over time within a subject) [5–7] which may be predictive of behavioral phenotypes. This growing body of work suggests that fMRI may hold

**Data Availability Statement:** All relevant raw data files are available from the Human Connectome Project (HCP) S1200 data database (https://www.

humanconnectome.org/study/hcp-young-adult/
document/1200-subjects-data-release). The
datasets underlying the results in this work are
provided in Supporting Information files. The codes
used in this study are available on Zenodo: Lee, K.
(2024). pyCAP codes: Human brain state dynamics
are highly reproducible and associated with neural
and behavioral features. Zenodo. https://doi.org/10.
5281/zenodo.13251562.

**Funding:** This work was supported by
5P50AA012870-22 SP-National Institute on
Alcohol Abuse and Alcoholism (NIAAA)/NIH/DHHS
(https://www.niaaa.nih.gov/) for J.H.K. and
National Institute of Mental Health (NIMH)/NIH/
DHHS (https://www.nimh.nih.gov/) grants
5U01MH121766-03 SP for A.A and P50MH109429
for J.D.M. L.B. was supported by the Fondation
Bettencourt Schueller (https://www.fondationbs.
org/en) and the Philippe Foundation (https://www.
philippefoundation.org/). G.R. was supported by
the Slovenian Research Agency (ARRS) grants P3-
0338, J7-8275, J5-4590 (http://www.aris-rs.si/en/
). The funders had no role in study design, data
collection and analysis, decision to publish, or
preparation of the manuscript.

**Competing interests:** K.L. consults for Manifest
Technologies. A.A. and J.D.M. hold equity with
Neumora Therapeutics (formerly BlackThorn
Therapeutics), Manifest Technologies, and are co-
inventors on the following patents: Anticevic A,
Murray JD, Ji JL: Systems and Methods for Neuro-
Behavioral Relationships in Dimensional Geometric
Embedding(N-BRIDGE), PCT International
Application No.PCT/US2119/022110, filed March
13, 2019 and Murray JD, Anticevic A, Martin WJ:
Methods and tools for detecting, diagnosing,
predicting, prognosticating, or treating a
neurobehavioral phenotype in a subject, U.S.
Application No.16/149,903, filed on October 2, 664
2018, U.S. Application for PCT International
Application No.18/054, 009 filed on October 2,
2018. J.L.J. is an employee of Manifest
Technologies, has previously worked for Neumora,
and is a co-inventor on the following patent:
Anticevic A, Murray JD, Ji JL: Systems and
Methods for Neuro-Behavioral Relationships in
Dimensional Geometric Embedding (N-BRIDGE),
PCT International Application No.PCT/US2119/
022110, filed March 13, 2019. C.F. consults for
Manifest Technologies and formerly consulted for
RBNC (formerly BlackThorn Therapeutics). G.R.
consults for and holds equity in Neumora and
Manifest Technologies. L.P. is an employee of
Manifest Technologies. J.H.K. holds equity in
Biohaven Pharmaceuticals, Biohaven
Pharmaceuticals Medical Sciences, Clearmind

great potential for characterizing how complex neural signals map onto human behavioral variation.

Spontaneous fluctuations of brain activity measured at rest (i.e., resting state functional Magnetic Resonance Imaging (rs-fMRI)) are embedded in time and space, exhibiting rich spatial-temporal information that varies within (state) and between (trait) individuals. The joint properties of state-trait rs-fMRI signal variation remains poorly understood, constituting a critical knowledge gap. An individual's mental state at any given time of rs-fMRI may be influenced by many intrinsic (e.g., metabolic) [8,9] or extrinsic (e.g., medications) factors that directly affect the circuit activity underlying complex behavior [10–18]. On the other hand, there might be other dimensions that contribute to variability in large neuroimaging data sets and undermine their ability to identify clear brain–behavior relationships. One of these dimensions may be time-varying signal dynamics. For example, personality theories posit that traits are characterized as patterns of thoughts, feeling, and behavior that generalize across similar situations within individuals and differ between individuals, whereas behavioral states reflect patterns that vary over time and situations [19,20].

Historically, rs-fMRI studies have quantified neural traits (e.g., stationary functional connectivity characterizing a subject) to study how they vary across people in relation to a given behavioral trait (e.g., fluid intelligence or a set of clinical symptoms) [21–23]. The analyses of neural state dynamics or time-varying rs-fMRI connectivity can be used to understand individual differences [24]. Evaluating moment-to-moment changes in neural activity can provide information about latent brain states associated with task-switching and decision-making in working memory [25]. Using dimension reduction of task fMRI data across multiple cognitive tasks, Shine and colleagues suggested that execution of diverse cognitive tasks and individual differences in fluid intelligence can be described using a dynamic flow along a low-dimensional manifold of global brain activity [26]. There is a knowledge gap regarding how combined state and trait variation of spontaneous brain dynamics map onto individual variation in complex behavioral phenotypes.

A recent meta-analysis of 3 large consortia data sets ($N = 38,863$ in total) has shown that brain–behavior associations in the general population have small effect sizes (e.g., $|r| < 0.2$) using data from thousands of individuals, when correlating neural measures from structural MRI, rs-fMRI, and task fMRI activation to behavioral measures including cognitive ability or psychopathology [27]. While large sample sizes are key for discovering and replicating small brain–behavior relationships on average [27], these recent advances leave the open question that there may be strong brain–behavioral effects that can be seen with quantitative approaches that consider time-varying signal dynamics [28–30]. Still, the application of state-related quantitative approaches in fMRI remain underutilized for characterizing reproducible inter-individual differences in brain–behavioral relationships [31,32]. Furthermore, combining state-related and trait-related information from rs-fMRI signals may provide convergent information about individual brain–behavior associations. To this end, we tested the hypothesis that reproducible neural-behavioral mapping may be achieved by quantifying combined state and trait information from time-varying rs-fMRI signals across the brain.

One approach that captures both trait and state neural characteristics is the analysis of co-activation patterns (CAPs) for rs-fMRI [33]. This analysis focuses on moment-to-moment changes in the whole brain blood oxygenation level dependent (BOLD) signals at each time point, providing a method to quantify the spatial patterns of co-activation across people and individual variation in patterns of neural temporal organization [33]. Several studies have reported similar average CAP patterns in healthy human adults [33], which also show some notable sex differences [34] and are impacted by proceeding task conditions [35]. Alterations of spatial and temporal organizations of CAPs (e.g., the number of time-frames occupied by a

Medicine, EpiVario, Neumora Therapeutics, Tempero Bio, Terran Biosciences, Tetricus, and Spring Care. J.H.K. consults for AE Research Foundation, Aptinyx, Biohaven Pharmaceuticals, Biogen, Bionomics, Limited (Australia), BioXcel Therapeutics, Boehringer Ingelheim International, Cerevel Therapeutics, Clearmind Medicine, Cybin IRL, Delix Therapeutics, Eisai, Enveric Biosciences, Epiodyne, EpiVario, Evidera, Freedom Biosciences, Janssen Research & Development, Jazz Pharmaceuticals, Leal Therapeutics, Neumora Therapeutics, Neurocrine Biosciences, Novartis Pharmaceuticals Corporation, Otsuka America Pharmaceutical, Perception Neuroscience, Praxis Precision Medicines, PsychoGenics, Spring Care, Sunovion Pharmaceuticals, Takeda Industries, Tempero Bio, Terran Biosciences, and Tetricus. All other co-authors declare no competing interests.

**Abbreviations:** ADP, Angular Deviation Penalty; BOLD, blood oxygenation level dependent; CAB-NP, Cole-Anticevic Brain Network Parcellation; CAP, co-activation pattern; CCA, canonical correlation approach; DT, dwell time; FD, frame displacement; fMRI, functional Magnetic Resonance Imaging; FO, fractional occupancy; FOV, field of view; HCP, Human Connectome Project; PCA, principal component analysis; rs-fMRI, resting state functional Magnetic Resonance Imaging; RT, reaction time; TMS, transcranial magnetic stimulation.

CAP state) were found across different levels of consciousness [36], schizophrenia [37], pre-psychosis [38], depression [39,40], and bipolar disorders [41,42]. All of these studies characterized group-level effects between patients and healthy controls with a fixed number of CAPs across groups, often capturing a parsimonious snapshot of brain dynamics by selecting a small number of time points associated with high-amplitude signals in preselected (i.e., seed) regions. While these studies have provided insights that CAPs contain rich information, they are systematically omitting full range of BOLD fluctuations. Put differently, few studies have leveraged the entire BOLD signal range to define CAPs [7]. Moreover, no study to our knowledge has investigated the properties of within and between-subject variability across a reproducible set of CAPs that harness the entire BOLD signal fluctuation range [43,44]. Finally, no study has in turn quantified how individual differences in CAP properties map onto complex behavior.

Here, we test the hypothesis that there is a reproducible CAP feature set that reflects both state and trait brain dynamics and that this feature set relates to individual phenotypes across multiple behavioral domains. To address this, we studied rs-fMRI and behavioral data obtained from 337 healthy young adults with no family relation in the individual Human Connectome Project (HCP) S1200 data [45]. To optimize neural features accounting for CAP variation within and between subjects, we develop a three-axes model of state-trait brain dynamics using moment-to-moment changes in brain CAPs. We identify 3 reproducible CAPs that can be quantified at the single-subject level, exhibiting recurrent snapshots of resting-state network spatial profiles and individual-specific temporal profiles. By analyzing spatiotemporal state-trait dynamics of CAP patterns, the data revealed groups of individuals that consistently exhibit behaviorally relevant CAP characteristics. These results suggest that a critical step toward the development of reproducible brain–behavioral models may involve initial mapping of neural features that can robustly and reproducibly capture combined trait (between-subject variability) and state (within-subject variability) variance in neural features.

## Results

### Three brain co-activation patterns are reproducibily found in healthy subjects at rest

The analysis of moment-to-moment changes in CAPs assumes a single neural state (i.e., CAP state) per each fMRI time frame and identifies a set of CAPs recurring over time and across subjects by spatial clustering of fMRI time frames [7,33].

We identify a reproducible set of CAPs from 4 runs of rs-fMRI data (15-min/run) obtained over 2 days from 337 healthy young adults (ages 22 to 37 years, 180 females) using a shuffled split-half resampling strategy across 1,000 permutations. Here, we used the entire BOLD signal fluctuation range for CAP estimations, without sparse time point sampling. In each permutation, we randomly split the sample ($N = 337$) into 2, each involving the equal number of non-overlapping subjects ($n = 168$, respectively, randomly excluding a subject) (**Figs 1A** and **S1**). To analyze CAPs at a low dimension space and to reduce the computational burden of CAP analysis that treats every 3D time frame in the clustering process (e.g., 4,000 time frames/subject), we used the Cole-Anticevic Brain Network Parcellation (CAB-NP) that involves 718 cortical surface and subcortical volumetric parcels [46]. We averaged the preprocessed BOLD signals in the voxels belonging to each parcel [47]. Therefore, within each split, a 4,000 x 718 array of individual rs-fMRI data are temporally concatenated across subjects. The time frames are clustered based on spatial similarity using K-means clustering, where the number of clusters ($k$) is estimated for each split using the elbow method varying $k$ from 2 to 15 (see the

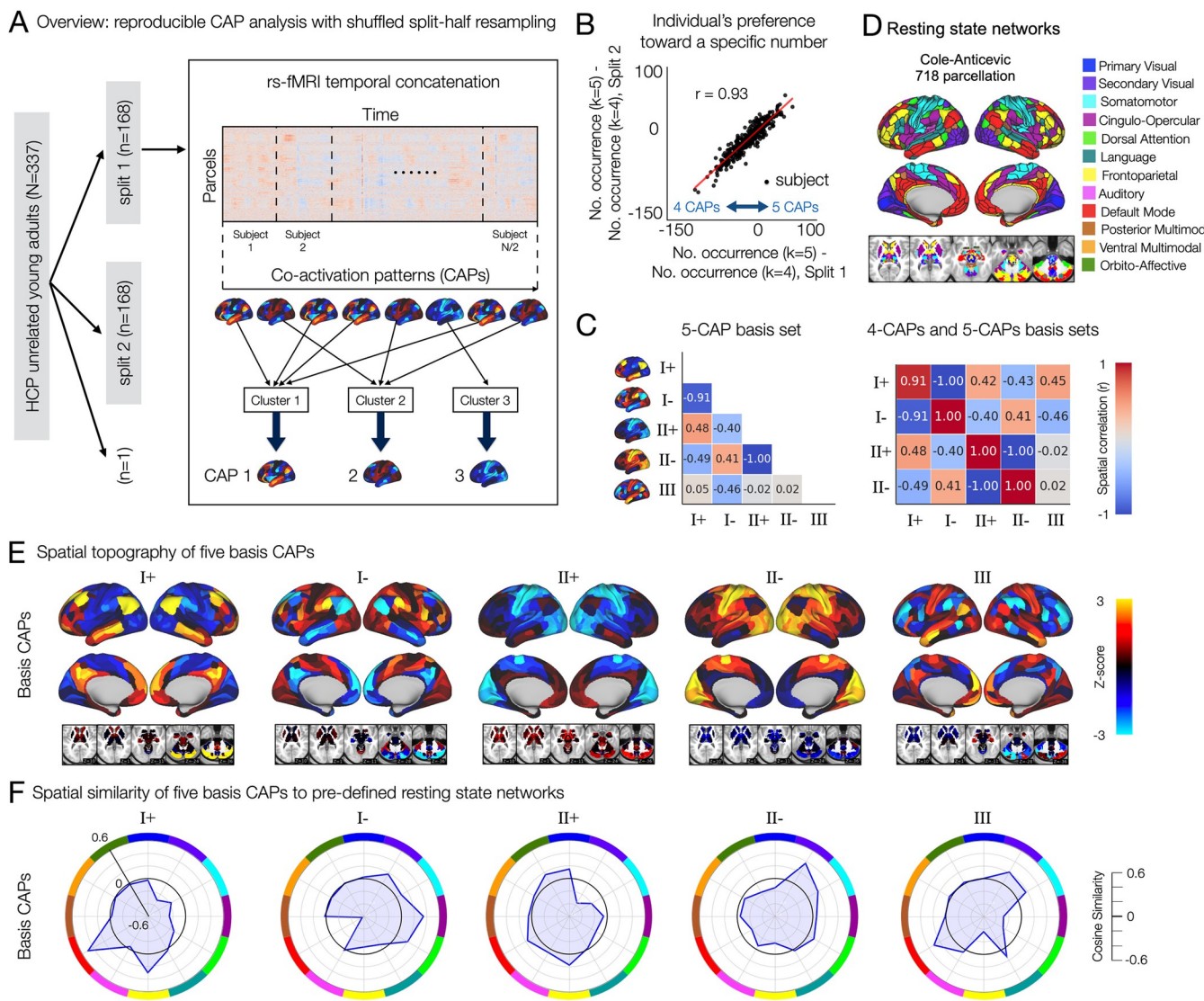

**Fig 1. A reproducible set of CAPs in the whole-brain rs-fMRI involve recurring mixed representations of canonical resting-state networks.** (**A**) Analysis overview. In each permutation, 337 subjects are randomly split into 2 equal-sized groups. Within each split, a parcel-by-time array of rs-fMRI data is temporally concatenated across subjects. Time frames are clustered based on spatial similarity using K-means clustering. The number of clusters (*k*) is estimated for each split. Each CAP is obtained as the centroid of each cluster (**S1 Fig**). (**B**) Individual's statistical preference toward a specific number of CAPs (*k*) is reproducible. In each split, an individual's preference toward a specific number was quantified using the number of permutations that resulted in a specific solution (e.g., 4 CAPs or 5 CAPs) across 1,000 permutations. Specifically, we compute the difference (occurrence of *k* = 5)—(occurrence of *k* = 4) for each subject (**Methods**). (**C**) Spatial correlation of the 5-CAP basis set (left) and between the 4-CAP basis set and the 5-CAP basis set (right). *r* values were rounded to the nearest 2 decimal digits. (**D**) CAB-NP [46]. (**E**) Spatial topography of 5 basis CAPs. (**F**) Spatial similarity of the 5 basis CAPs to canonical resting-state networks, predefined using the CAB-NP parcellation in (**D**). CAB-NP, Cole-Anticevic Brain Network Parcellation; CAP, co-activation pattern; rs-fMRI, resting state functional Magnetic Resonance Imaging.

estimated Silhouette scores from the K-means clustering solutions in **S2 Fig**). Finally, a CAP was obtained by averaging the time frames within each cluster with respect to each parcel.

We first found that there are individual differences in the number of reproducible brain states. Specifically, in both splits, the estimated number of CAPs was either 4 or 5, each exhibiting an ≈50% occurrence rate across permutations (**S3A and S3B Fig**). However, interestingly, the co-occurrence of the same number of CAPs in both splits was rare (<6%) (**S3C Fig**). In other words, a half of the sample produced 5 CAPs, while the other half produced 4 CAPs.

Because each of 2 nonoverlapping halves contain a distinct subset of samples, we hypothesized that individual difference in the number of reproducible brain states plays a role in the observed between-split differences. To test this hypothesis, we quantified the individual's preference toward a specific number of CAPs by comparing the probability of estimating 4 CAPs or 5 CAPs. The probability of estimating $k$ CAPs was quantified using the occurrence of $k$ solution estimations in a split across permutations (see **Methods**). Indeed, there was a highly reproducible tendency for individual subjects to occupy either 4 or 5 CAPs (**Fig 1B**). Together, these results suggest the presence of a CAP state that is reproducibly found in a subset of subjects but not in others.

To identify reproducible spatial topography of CAPs for further analyses, we generated 2 sets of basis CAPs independently: the 4-CAP and the 5-CAP basis sets (**S4 Fig**). The 4-CAP basis set was obtained by applying agglomerative hierarchical clustering to the CAPs collected from only the permutations that resulted in the estimation of 4 CAPs. Then, a basis CAP was generated by averaging the CAPs belonging to each cluster, and the value in each parcel of the basis CAP was normalized to z-scores using the mean and standard deviation across 718 parcels (**S4 Fig**). The 5-CAP basis set was also obtained using the CAPs collected from the permutations resulting in 5-CAP solutions. We found that the 4-CAP basis set consisted of 2 pairs of anti-correlated CAPs (I+ and I-, II+ and II-), and the 5-CAP basis set consisted of the same 2 pairs of anti-correlated CAPs and 1 additional CAP (III) (**Fig 1C**). The patterns of these basis CAPs were consistent between 2 splits (**S5 Fig**). The number (I, II, and III) and sign (+ and −) of CAPs were labeled arbitrarily. Overall, we found 3 CAPs recurring over time and across healthy subjects in rs-fMRI.

## Patterns of whole-brain co-activation are recurrent snapshots of mixed resting-state networks

As expected, the spatial patterns of 3 CAPs were associated with well-known rs-fMRI networks (**Fig 1E and 1F**). CAP I involved a strong bi-polarity between the default mode and frontoparietal networks versus the dorsal attention, cingulo-opercular, somatomotor, and secondary visual networks. Here, bi-polarity stands for positive versus negative cosine similarity of each CAP with distinct resting-state networks (CAP+ versus CAP−). CAP II exhibited a weaker bi-polarity between the primary visual, orbito-affective, default mode, and frontoparietal networks versus the dorsal attention, somatomotor, and secondary visual networks. CAP III showed a strong bi-polarity between the default mode, somatomotor, and secondary visual networks versus the frontoparietal, dorsal attention, and cingulo-opercular networks. Considering that resting-state networks are identified based on the co-fluctuations of signals in distributed brain regions, our results show that these CAPs represent recurring snapshots of the diverse signal co-fluctuations among regions involved in different functional networks at each time frame.

## CAP III is reproducibly found in some individuals but not in others

Our result in **Fig 1B** suggests that there are individual differences in the number of reproducible brain states. Because CAPs are estimated using data from a group of subjects, the contribution of a single subject to this estimation is relatively small. In addition, it remains unknown whether the spatial topography of estimated CAPs are reproducible across permutations. To address these, we investigated three questions: (i) whenever 4 CAPs are estimated from a split data, are their spatial patterns reproducible across the permutations; (ii) whenever 5 CAPs are estimated from a split data, are their spatial patterns reproducible across the permutations; and (iii) is there a specific CAP state that is reproducibly missing in 4-CAP solutions when

compared to the 5-CAP solutions. All 718 cortical and subcortical parcels were included in this and following analyses throughout this article.

First, we calculated the marginal distribution of spatial correlation values ($r(EC_i, BC_j)$) between the CAPs estimated from each split data (Estimated CAP; $EC_i$, $i$ = 1,.., 4 or 5) and a given basis CAP (Basis CAP; $BC$) (**Fig 2A**). Note that these predefined basis CAPs are the group-average and permutation-average CAPs obtained using the agglomerative hierarchical clustering of all CAPs across permutations (**Fig 1E**). In each permutation, each $EC_i$ was labeled according to the maximum rank correlation with the given basis CAP. As a result, the marginal distribution of $r$ values showed that the spatial patterns of 4-CAP solutions and 5-CAP solutions were strongly reproducible (**S6 Fig**). The CAPs estimated from each split were highly correlated with at least one of the basis CAPs, demonstrating a 1-on-1 matching for all CAPs. In addition, CAP III was reproducibly found in one split but not in another split across permutations (**Figs 2** and **S7**). Together, this analysis demonstrates that the presence or absence of CAP III is not a random artifact but actually associated with reproducible neural dynamics of individuals.

## Spatial alignment of individual time frames to basis CAPs

To find an optimal number of clusters or CAPs that are commonly found across individuals, we used an approach that considers a trade-off between the number of clusters and within-cluster similarity by combining the silhouette criteria and elbow method (**S2 Fig**). To evaluate the extent of the contribution of individual co-activation patterns to the observed CAP variability, we analyzed all fMRI time frames obtained from 337 subjects after scrubbing. For each split, we computed the spatial alignment of individual 3D fMRI time frames to the 5 basis CAPs (cluster centroids estimated by K-means clustering) using Pearson's correlation, identifying a basis CAP yielding the highest correlation with each time frame. As a result, the mean and standard deviation of the maximum correlation were 0.22±0.11 (**S8 Fig**), indicating the substantial variability in resting state human brain dynamics. Notably, the group-level spatial topography of CAPs, estimated by averaging the time frames within each cluster, remained consistent across permutations (**Fig 2**), enabling us to investigate individual differences in their temporal dynamics.

## Reproducible state-trait neural features at the single-subject level

We identified 3 CAPs that reflect brain-wide motifs of time-varying neural activity. Here, we demonstrate a reproducible estimation of spatial CAP features at the single-subject level. The CAP analysis involves the assignment of individual time frames to one of the estimated CAPs using the K-means clustering process (**Fig 3A**). The CAPs estimated in each split were labeled using the maximum ranked correlation with the pre-identified 5-CAP basis set (**S6 Fig**). In turn, this frame-wise identification of CAP states allows the estimation of temporal profiles of CAP states for individual subjects. We demonstrate that reproducible state and trait features of neural dynamics can be quantified using several key parameters of CAP temporal characteristics (see **Fig 3A**).

## Definitions

1. *Fractional occupancy* (FO($s$, $i$)): the total number of time frames (or MRI time of repetition; TR) that a subject $s$ spends in CAP state $i$ per day, normalized by the total number of time frames spent in any CAP state by subject $s$ per day. FO is a relative measure (%$TR$),

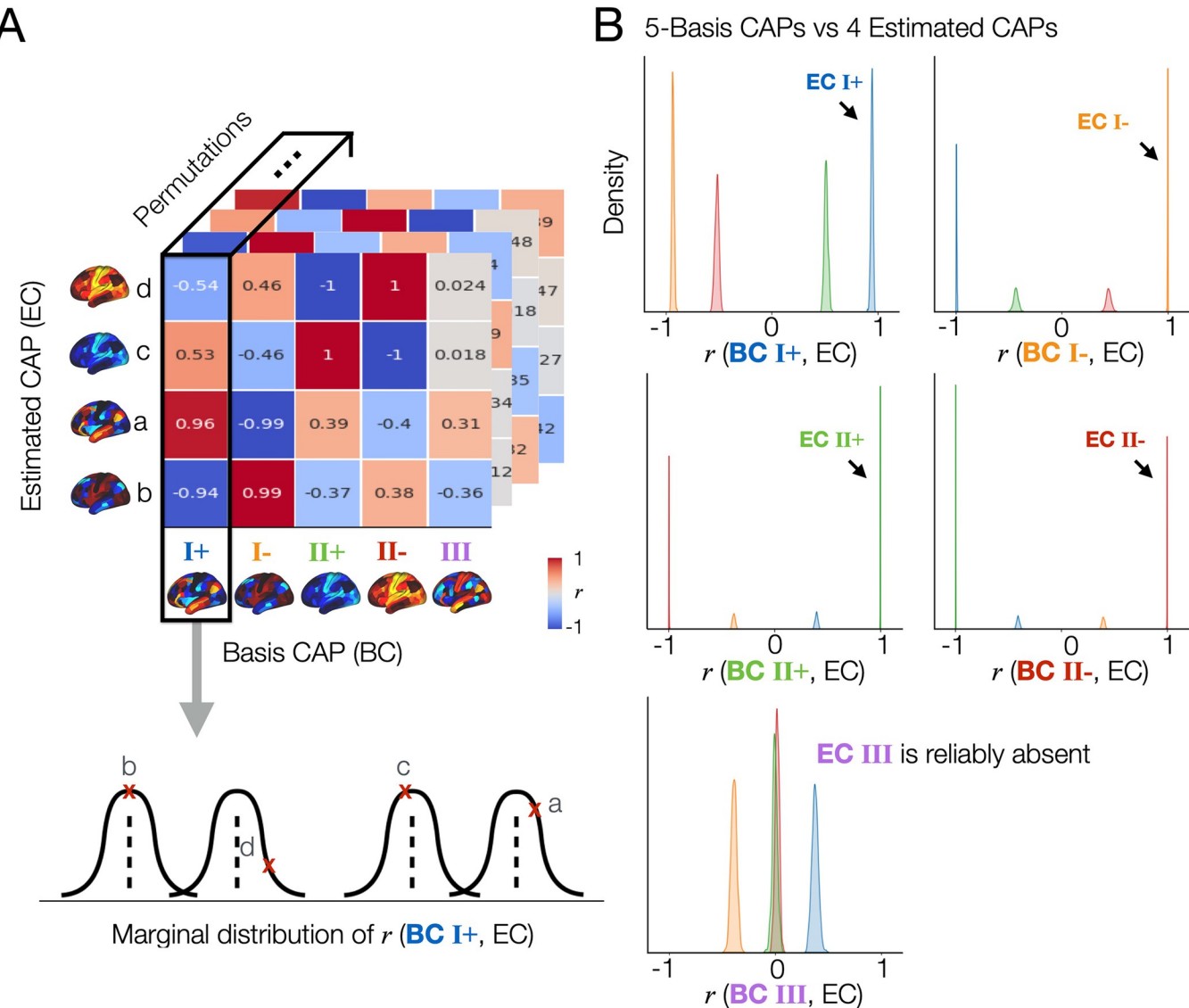

**Fig 2. The spatial patterns of the CAPs estimated across split-half permutations are reproducible, demonstrating the consistent absence of a specific spatial pattern (CAP III) in one split but not in another split across permutations. (A)** Proof of concept. In this figure, we demonstrate "whenever 4 CAPs are estimated from a split data, are their spatial patterns reproducible across the permutations," and "if there is a specific CAP state that is reproducibly missing in 4-CAP solutions when compared to the 5-CAP solutions." To address these, first, among 1,000 permutations, we only take permutations that resulted in 4-CAP solutions using the elbow method, which was 502 permutations in this data. The remaining 498 permutations mostly resulted in 5-CAP solutions, and rarely 6- or 7-CAP solutions as shown in (S3 Fig). Spatial similarity ($r$, correlation coefficient) is computed between each of the estimated CAPs (EC; denoted as a, b, c, and d) and a given basis CAP (BC). In this example, we select BC 1 from the 5-CAP basis set. $r$ values were rounded to the nearest 2 decimal digits for visualization. Finally, we obtain the marginal distribution of $r$ values between BC 1 and the estimated CAPs across 502 permutations. **(B)** The CAP III is reproducibly found in the 5-CAP solutions and not in the 4-CAP solutions across permutations. We repeated the spatial similarity analysis for the 4 CAPs estimated from each split-half data, when compared to the 5-CAP basis set. In each permutation, each estimated CAP was labeled according to the maximum rank correlation with the basis CAPs. Data-points ($r$-values) estimated from the CAPs with a same label were coded using the same color. The marginal distributions of $r$ between all estimated CAPs and each BC from the 5-CAP basis set are illustrated using kernel density estimation. Results obtained from the split 1 data are shown in (**B**) and replicated in the split 2 data (see S7 Fig). Note that all 718 cortical and subcortical parcels were included in this analysis. For simplicity, subcortical regions of CAPs are not visualized. CAP, co-activation pattern.

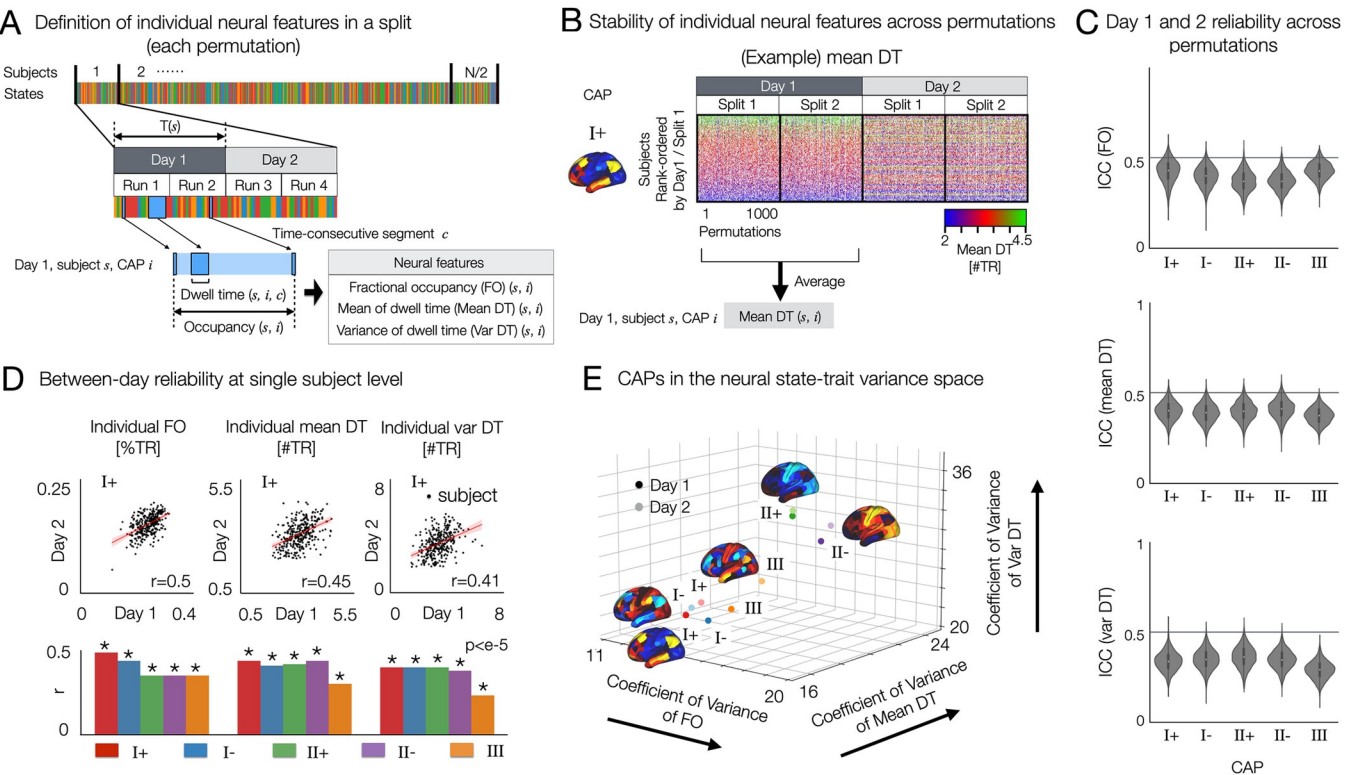

**Fig 3. Resting state brain CAPs have distinct between and within-subject variance of temporal characteristics and test-retest reliability, as revealed by the 3-axes representation of neural trait variance space. (A)** Analysis overview. In each split-half data from each permutation per day, FO, within-subject mean of dwell time (Mean DT) and within-subject standard deviation of dwell time (Var DT) are estimated for each CAP state. **(B)** Stability of individual mean DT of CAP I+ across permutations and across 2 days. Individual subjects were rank-ordered from top to bottom using the split 1 data from day 1. While the estimated mean DT values spanned from 0 to 6, the data set exhibited sparse occurrences in the distribution tails. To enhance visual clarity across rows (subjects), a saturated colormap was employed. For an alternative representation of the same data using an unsaturated colormap, refer to **S9 Fig**. We also found that individual Var DT and individual FO for these CAPs are reproducible across permutations and 2 days (**S9 Fig**). **(C)** Days 1 and 2 reliability of FO (top), Mean DT (middle), and Var DT (bottom) in each CAP state were quantified by the intraclass correlation coefficient using two-way random effect models (ICC(2,1)). When computing ICC for CAP III, permutations resulting in the absence of CAP III was not considered, because the values of temporal metrics are zero for both days. **(D)** Test-retest reliability of neural measures between 2 days of scan. (Top) For CAP I+ state, we show scatter plots of individual FO, within-subject mean and variance of DT between days 1 and 2. Linear fitting line (red) is shown for each scatter plot. *r*-value is measured by Pearson's correlation coefficient and considered significant when the corresponding two-sided *p*-value is less than 0.001. (Bottom) For the remaining 4 CAP states, the same scatter plot analysis was repeated (**S10 Fig**). We summarize the estimated *r*-values from all CAPs in the bar plot. **(E)** CAPs on the neural trait variance space. Relative variance (coefficient of variance) of each CAP measure was computed across subjects: individual FO (*x*-axis), Mean DT (*y*-axis), and Var DT (*z*-axis). The three-axes representation allows for unifying and optimizing the variations of temporal CAP characteristics and distinct patterns of temporal organizations of brain activity. Note that all 718 cortical and subcortical parcels were included in this analysis. See subcortical regions of CAPs in **Fig 1E**. The data used to generate the results can be found in **S1 Data**. CAP, co-activation pattern; DT, dwell time; FO, fractional occupancy.

such that the sum of FO of all CAP states is 1 within a subject per day. FO reflects between-subject variance (trait variance) of CAP dynamics.

2. ***Dwell time*** (DT(*s*, *i*, *c*)): the number of time-frames (#*TR*) of a time-consecutive segment *c* occupying the same CAP state *i* within a subject *s* per day.

3. ***Within-subject mean of DT*** (Mean DT(*s*, *i*)): the mean of estimated values of DT for all time-consecutive segments during which CAP *i* is occupied by subject *s* per day.

4. ***Within-subject variance of DT*** (Var DT(*s*, *i*)): the standard deviation of estimated values of DT from all time-consecutive segments occupying a CAP *i* within a subject *s* per day. DT measures involve both trait (between-subject) and state (within-subject) components of neural dynamics.

The quantification of these CAP measures was performed for each split data per permutation. To evaluate day-to-day variability of CAP dynamics, we computed these measures for each day separately. In summary, we estimated FO, mean DT, and var DT for each CAP per subject. This allowed us to average the estimated neural measures across permutations, providing a summary statistic of neural measures for each CAP for each subject per day. These statistics are statistically reproducible at the single-subject level, as shown in **Fig 3B** [48–50]. Care is needed when interpreting the results, because stable individual-specific properties of state dynamics such as mean DT in this study can also be considered as traits.

In this study, we are interested in testing the hypothesis that there is a reproducible general motif of individual differences in neural co-activation dynamics, where individuals differently occupy (or project onto). While previous work in [26] focused on a low-dimensional manifold of spatiotemporal neural activity by applying principal component analysis (PCA) of rs-fMRI signal volumes, we aim to identify a low-dimensional feature space that characterizes state and trait properties of the temporal organizations of brain states. To do this, we first demonstrate that state-trait CAP dynamics are reproducible at the single-subject level across permutations, whereas within-subject between-day reliability was lower than between-permutation reliability on a same day (**Figs 3B, 3C** and **S10**). First, we measured the test-retest reliability of the neural measures using a linear regression (**Fig 3D**). For each CAP, we found a moderate low correlation ($r \leq 0.5$) of individual neural measures between day 1 and day 2 (**Fig 3D**). CAP I+ showed the highest between-day reliability and CAP III was the lowest. See **S10 Fig** for the scatter plots from the other 4 CAP states. When calculating the mean and SD of correlation across all CAPs, the between-day correlation is 0.41±0.07 for FO, 0.41±0.06 for mean DT, and 0.38±0.07 for var DT.

Secondly, we computed the intraclass correlation coefficients using two-way random effect models (ICC(2,1)) for each split in each permutation. Therefore, for each CAP, we measure 2,000 ICC values across 1,000 permutations. The average ICC across all CAPs are 0.39±0.06 (mean ± standard deviation) for FO, 0.39±0.05 for Mean DT, and 0.34±0.06 for Var DT. These state-trait neural measures show fair test-retest (day-to-day) reliability, when compared to the meta-analytic estimate of average ICC (0.29±0.03, mean ± standard error) across other studies reported using edge-level functional connectivity [51]. Within-subject variance of FO across 5 CAPs are shown in **S12 Fig** across permutations. Together, these results show day-to-day variability (state) in CAP dynamics within individuals and highly reproducible between-subject (trait) variability within each day.

## Joint analysis of state and trait neural variations

We propose an analytic framework of joint state and trait neural variations, taking the test-retest (or day-to-day) reliability of neural features into account. Importantly, this framework allows us to visualize how CAP properties that vary within a person (state) also vary between people (trait). In **Fig 3E**, we illustrate a three-axes representation of state and trait variance components of spatiotemporal CAP dynamics. For each CAP, we estimate the normalized inter-subject variance (coefficient of variance) of 3 neural features. Then, the 5 CAP states (CAPs I+/−, II+/−, and III) are projected on this space. Interestingly, we found that CAP II exhibits the highest relative between-subject variation (i.e., trait) across all 3 measures, the FO, mean DT, and var DT. Conversely, CAP III exhibits lower between-subject variance but higher within-subject variance than CAP II (as seen in the distance between the measures on 2 different days; see **Fig 3E**). Indeed, the proposed joint analysis of state-trait neural variations provides a rich landscape of within-person and between-person variance of neural co-activations.

## Neural feature reduction captures general motifs of individual variation

An important and interesting question would be whether neural features with distinct patterns of state-trait variation can provide vital information about individual differences. Put differently, we are interested in studying if there is a set of neural features that can be commonly found across a number of healthy subjects that have a reproducible set of neural co-activation properties, which can in turn be related to behavioral phenotypes. To address this question, we first collected 30 neural features estimated for each individual: 3 neural measures (FO, mean DT, and var DT) × 5 CAPs (I+, I-, II+, II-, and III) × 2 days. We performed the agglomerative hierarchical clustering of a subject-by-feature (3.37×30) matrix (**Fig 4A**). We determined the number of clusters using a distance cut-off value of 70% of the final merge in the dendrogram (**Fig 4B**). As a result, we found 3 subgroups (**A**, **B**, and **C**), each consisting of 163, 127, and 47 individuals (**Fig 4C**).

To further study if there is a low-dimensional geometry of neural state-trait variation capturing individual differences, we applied PCA to the subject-by-feature matrix. Clearly, the 3 subgroups identified using hierarchical clustering were distributed in the low-dimensional space represented by the first 3 neural PCs, which explain 33.5%, 23.9%, and 16% of variance, respectively (**Fig 4D**). Notably, subgroup **A** shows higher scores on neural PC 1 than the other groups, and subgroup **C** shows higher scores on neural PC 2 than subgroup **B** (**Fig 4C**). Our further analysis of feature loadings on these PCs revealed a unique and reduced feature set of neural variation, each representing CAP-specific (PC 1) and general (PC 2) neural state-trait variations, which also exhibit day-to-day variability (PC 3). In addition, we found that each pair of positive and negative CAP patterns (states I+ and I-, states II+ and II-) exhibit similar temporal CAP profiles (**Figs 4E** and **S11**).

Specifically, the neural PC 1 is characterized by distinct temporal profiles on CAPs I/III versus CAP II. It includes higher loadings of FO, mean DT and var DT at CAPs I/III and lower loadings of DT measures at CAP II (**Fig 4F**). Note that the FO is a relative measure (#*TR*) such that the sum of FO at all CAP states is 1, whereas the DT measures are absolute (#*TR*). This indicates that individuals exhibiting high scores on neural PC 1 occupy CAPs I and III for a relatively longer time, whereas individuals with low PC 1 scores occupy CAP II state for a longer time. Regarding CAP II, the FO exhibits a more pronounced negative loading on neural PC 1 compared to the dwell time measures (mean DT and var DT). On the other hand, the neural PC 2 highlights a general pattern of state persistence (high within-subject mean DT and high within-subject variance of DT), while also exhibiting a weak CAP-specific effect on FO (lower loadings of the FO at CAPs I/III and higher loadings of FO at CAP II) (**Fig 4F**). In addition, in neural PC 2, the DT measures of CAP II showed higher loadings than FO. A lengthy dwell time indicates that an individual occupies a state for an extended duration before transitioning to another CAP, suggesting strong state persistence. In contrast to the neural PCs 1 and 2 that showed strong between-day reliability, neural PC 3 showed a strong negative correlation between days |r| > 0.9; **Fig 4G**). In particular, neural PC 3 captures a specific component of day-to-day variability: the CAP-specific patterns observed in neural PC 1 can undergo systematic changes between days (e.g., sign-flipped feature loadings in **Fig 4F**).

Together, our results demonstrate that both state and trait variance of spatiotemporal CAP dynamics involve pivotal information for identifying individual differences. Specifically, we identified 3 neural PCs that establish a low-dimensional, general motif of state and trait neural co-activation variation. The third principal component of individual variation involved information about day-to-day variability in neural co-activation, suggesting that patterns of within-subject variations can be uniquely individualized. This can be, in turn, considered as trait-like patterns providing additional information about individual neuro-phenotypes. While trait

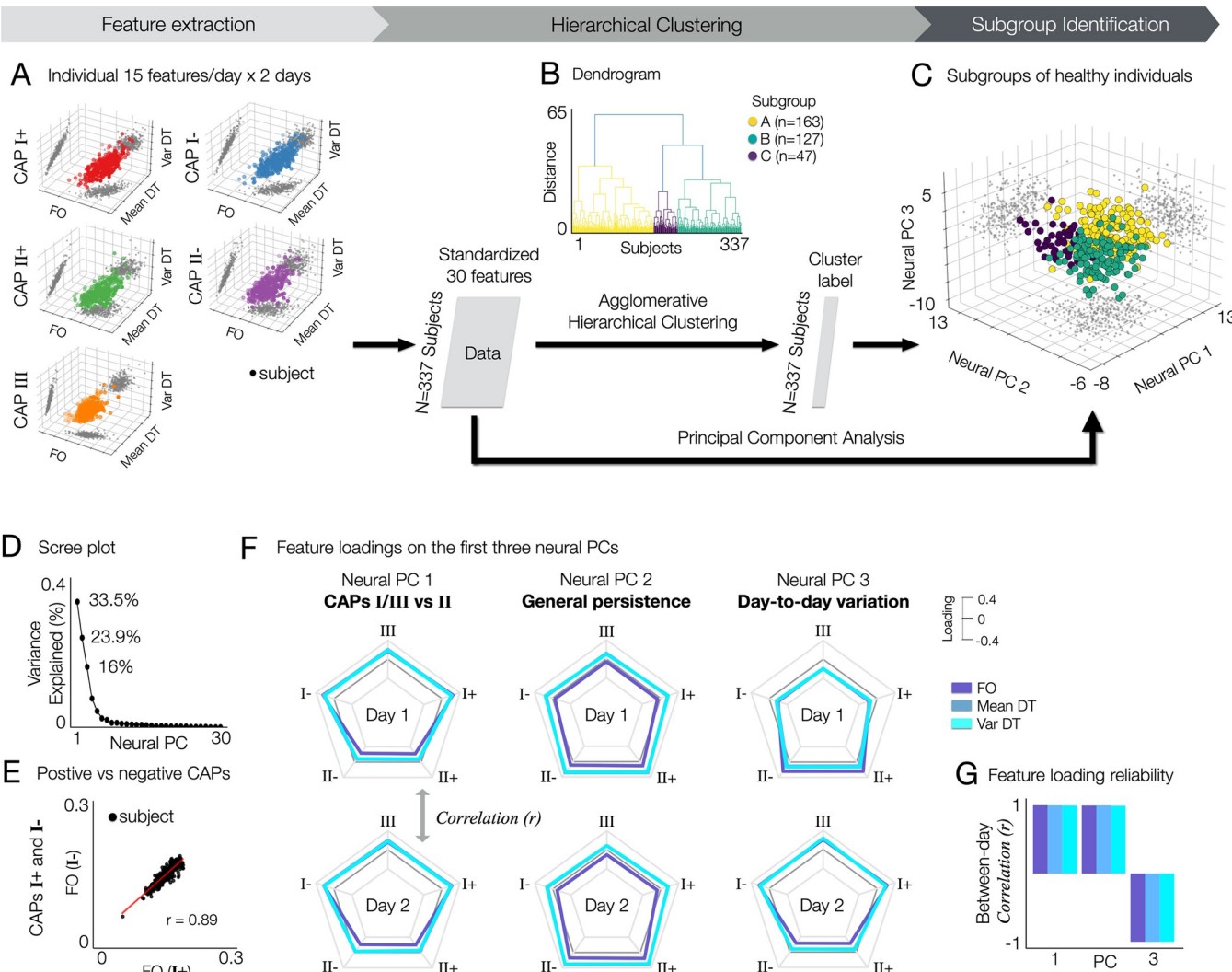

**Fig 4. Identification of subgroups in healthy subjects exhibiting distinct neural state-trait variances.** Three subgroups of healthy subjects in the HCP data (**A**, **B**, and **C**) are identified using the agglomerative hierarchical clustering of 30 individual neural state-trait features, which are estimated from temporal CAP characteristics (fractional occupancy, FO; within-subject mean of dwell time, mean DT; within-subject variance of dwell time, var DT). (**A**) For each subject, 30 neural features estimated from 5 CAPs and 2 days are collected. For each CAP, each neural feature was obtained by averaging the values estimated across permutations. Each data-point in the 3-axis scatter plots indicate a subject. Individual neural features were obtained by averaging the feature values across permutations within subject for each day. (**B**) Agglomerative hierarchical clustering is performed on the feature matrix. In the dendrogram, 3 clusters are found using a distance cut-off value of 7% of the final merge. In addition, to estimate the principal geometry of this state-trait feature space identifying subgroups, we applied PCA to the feature matrix. (**C**) Clustered subjects are embedded onto a 3D space using PCA. (**D**) Variance explained (%) by each neural PC. (**E**) Similarity of individual neural features between positive and negative CAPs. An example of CAPs I+ and I- are shown. See **S11 Fig** for all results (0.9±0.04, mean ± SD). (**F**) Loadings of each neural feature on the first 3 neural PCs. In each radar plot, 3 lines indicating FO (colored in slateblue), Mean DT (steelblue), and Var DT (turquoise) are shown for 5 CAPs. Feature loadings from days 1 (top) and 2 (bottom) are shown separately for an easier interpretation, while the neural PCs were obtained using neural features from both days as shown in (**A**). (**G**) The loadings of neural features on each PC are reliable between days. For each neural PC, Pearson's correlation coefficient (r) was computed between 2 vectors of feature loadings collected from days 1 and 2. Neural PC 3 reflects the contribution of within-subject (between-day) variance in temporal CAP profiles. CAP, co-activation pattern; DT, dwell time; FO, fractional occupancy; HCP, Human Connectome Project; PCA, principal component analysis.

variations (neural PC 1 and 2) are dominantly loaded on the general motif of individual differences, the observed state variance at the time scale of days (neural PC 3) also contributes to this low-dimensional feature space, therefore reflecting neuro-phenotypes. The assessment of individual distributions of each neural measure supported these findings (**S13 Fig**). In

addition, we found that the FO of CAPs I and II have overall a higher mean and variability than the FO of CAP III. We observed the same patterns in the mean DT and var DT (**S13 Fig**). Indeed, our analyses combining the hierarchical clustering and PCA of individual neural feature sets revealed 3 subgroups exhibiting distinct patterns of neural variations.

## Principal variations of neural state-trait features co-vary with principal variations of behavioral phenotypes

Next, we were interested in studying to what extent individual variability quantified on this low-dimensional neural feature space was linked to variations of individual human behavior. We employed a similar dimension reduction strategy to estimate the behavioral principal components (PCs) that provide low-dimensional geometries across multiple behavioral domains where people occupy differently. This way, we can associate how individual subjects are distributed in 2 feature spaces respectively and how such patterns relate to each other.

To estimate the geometry of principal variations in behavioral phenotypes, we performed PCA on 262 variables across 15 behavioral domains from the HCP S1200 unrestricted and restricted behavioral data: alertness (1–2), cognition (3–39), emotion (40–63), personality (64–68), emotion task performances (69–74), gambling task performances (75–86), language task performances (87–94), relational task performances (95–100), social task performances (101–113), working memory task performances (114–167), psychiatric dimensions (168–189), alcohol use (190–222), tobacco use (223–252), illicit drug use (253–258), and marijuana use (259–262) (**Fig 5A**). Find the list of behavioral variables in **S14 Fig**. Before performing PCA, several variables reflecting the reaction time (RT) in tasks were converted to 1/RT for a better interpretation of PC geometry.

After performing PCA, the significance of derived PCs was evaluated using permutation testing. Specifically, PCA was performed for each permutation where the order of subjects was randomly shuffled, which in turn provided a null model [23]. As a result, we found 27 PCs that accounted for a proportion of variance that exceeded chance ($p<0.05$ across 10,000 permutations). Subsequently, we considered the first 15 PCs, which collectively explained approximately 50% of the total variance, for further analyses. Reproducibility of these 15 PCs was evaluated using a split-half permutation approach, where we randomly splitted 337 subjects into 2 equal sized groups ($n = 168$) and applied PCA for each split. Then, the similarity (Pearson's correlation) of PC geometry between the $n$-th PCs estimated from 2 split-halves was computed for each permutation, where $n$ is the ranked order of each PC based on explained variance.

As a result, we found that the first behavioral PC (PC 1) explaining 11.2% of variance (**Fig 5B**) was highly reproducible, exhibiting the similarity ($r = 0.9\pm0.03$, mean ± SD across 1,000 permutations) of PC geometry between the first PCs estimated from 2 split-halves (**Fig 5C**). The behavioral PC 1 highlighted individual life function outcomes associated with cognitive function, emotion regulation, and alcohol and substance use (**Figs 5D** and **6A**). The variables of working memory task performances have the highest loadings on the behavioral PC 1, followed by the emotion, relational, languages, gambling task performances, fluid intelligence, self-regulation/impulsivity, and episodic memory. In contrast, variables associated with alcohol and substance use (e.g., short-term tobacco use) and psychiatric dimensions (e.g., self-report measures of positive and negative affect, stress, anxiety, depression, and social support) exhibited the lowest, negative loadings on the behavioral PC 1. Behavioral PC 2 highlighted items associated with emotion, personality and psychiatric life functions (**Fig 6B**). Behavioral PC 3 highlighted substance use, showing notable high loadings of alcohol consumption habit-related items on this PC (**Fig 6C**).

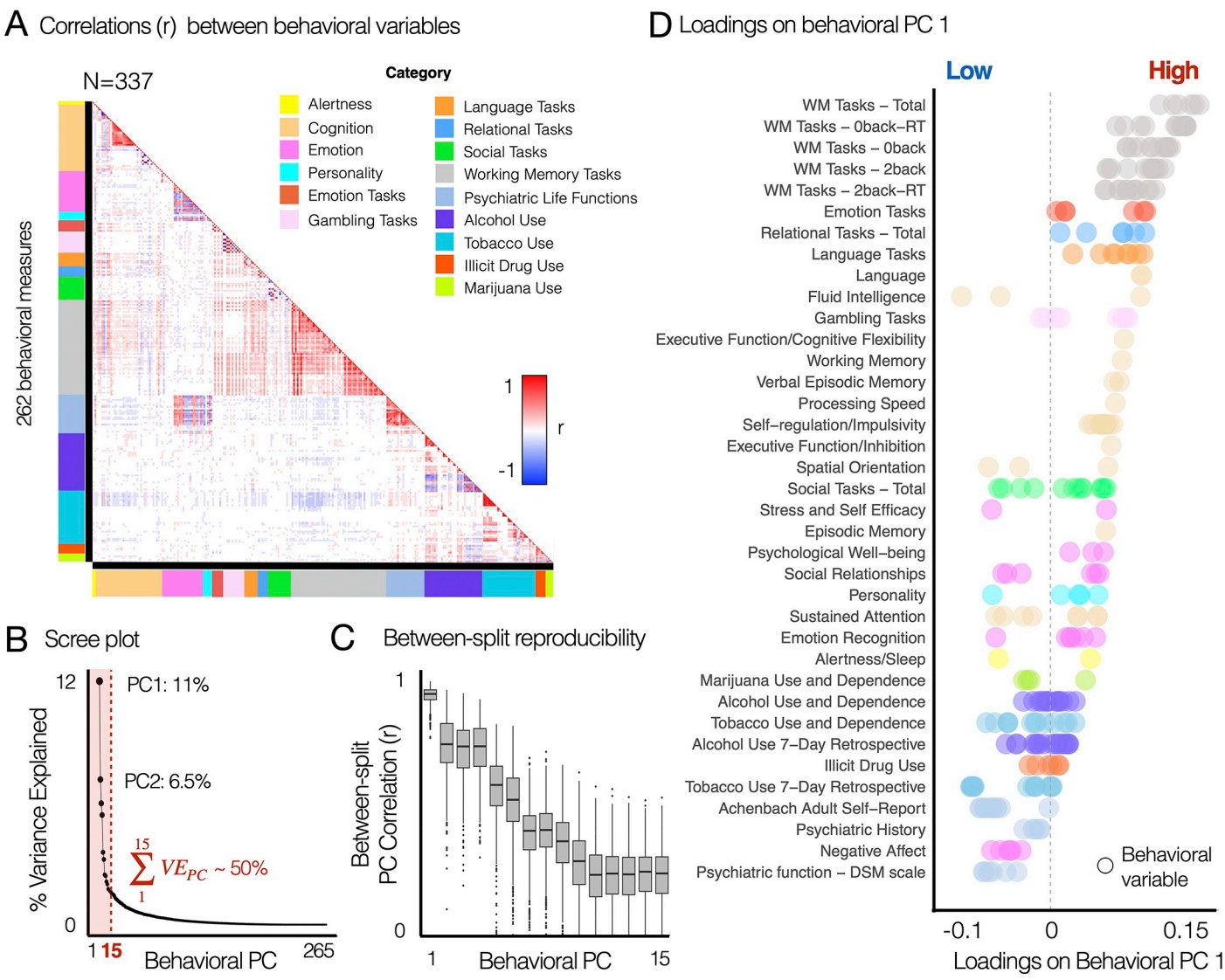

**Fig 5. Principal variations of neural state-trait features co-vary with the principal variations of behavioral phenotypes, highlighting individual life function outcomes associated with emotion regulation, cognitive function, and alcohol and substance use.** (A) Correlation structure between 262 behavioral variables, which were obtained from the HCP S1200 unrestricted and restricted data. Colorbars along each axis of the correlation matrix indicate color-codes for the category of each variable. Categories were defined from the HCP data dictionary available online (HCP_S1200_DataDictionary_April_20_2018.csv). Variables measuring RT from tasks were transformed into 1/RT to account for the fact that a shorter response time indicates better task performance. See S14 Fig for the list of all behavioral variables. (B) The first PC explained 11.2% of variance. The first 15 PCs explaining ~50% of variance were considered in further analysis. (C) Across 1,000 permutations for split-half resampling, we compared if the geometry of estimated PCs in 2 splits are consistent. Pearson's correlation coefficient ($r$) was computed for each pair of behavioral PCs. (D) Rank-ordered loadings of each behavioral variable on the first principal component (PCA). Each data point indicates a behavioral variables. PCA was performed for all 262 variables in (A). 39 subcategories shown on the y-axis were also defined using the HCP data dictionary. Several subcategories belonging to the same category are coded using the same color as in (A). The data used to generate the results can be found in S2 Data. HCP, Human Connectome Project; PCA, principal component analysis; RT, reaction time.

To assess the association between the principal variation of behavioral variables and the principal variations of neural features, we first compared the distribution of individual scores on 15 behavioral PCs between the subgroups, identified using the neural features (**Fig 4**). Individuals classified as subgroup **A** ($n = 163$) exhibited significantly higher scores on behavioral PC 1 compared to subgroup **B** ($n = 127$) ($p_{BON} < 0.05$, $t = 3.05$, two-sample two-sided $t$ tests)

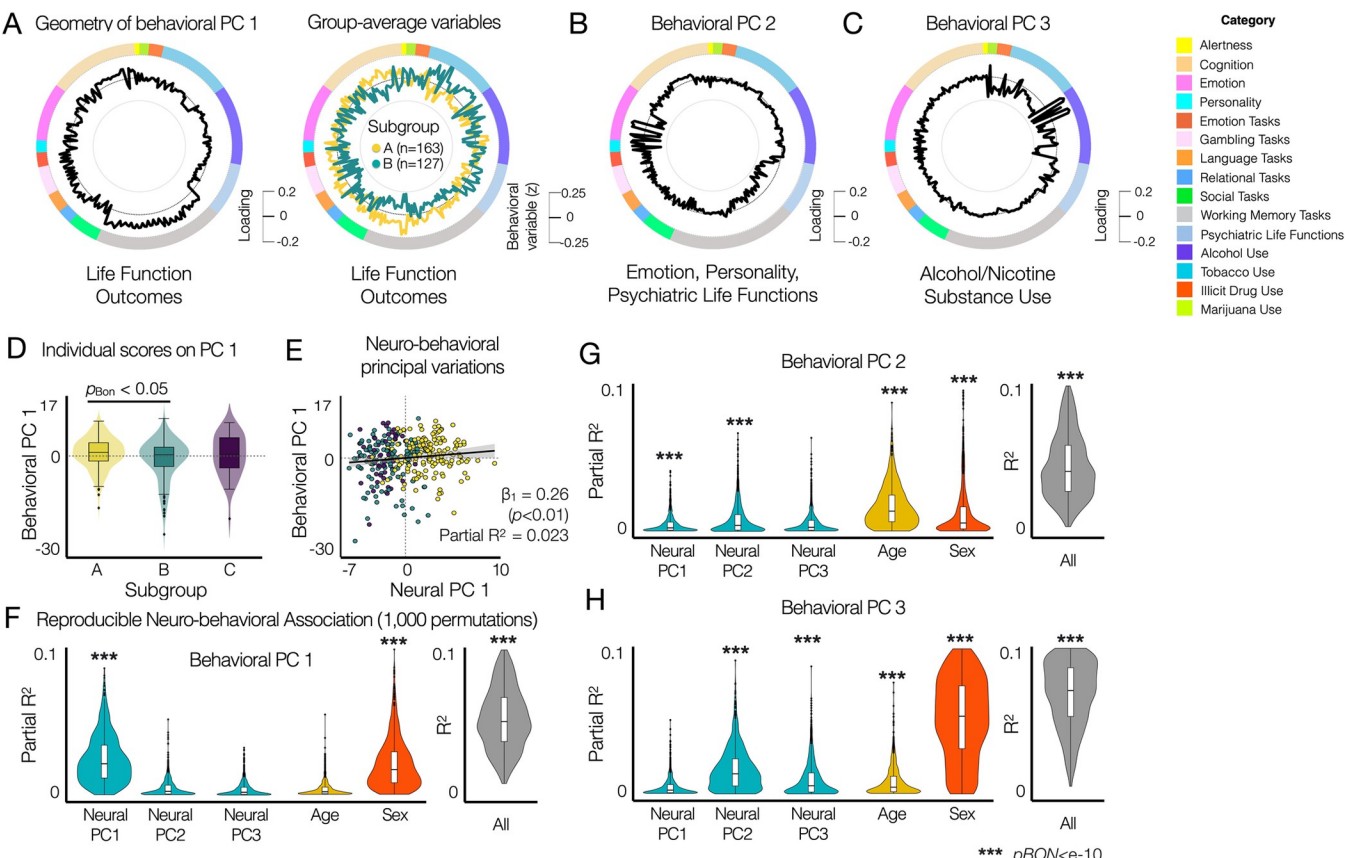

**Fig 6. Principal variations of neural state-trait features co-vary with the principal variations of behavioral phenotypes, highlighting individual life function outcomes associated with emotion regulation, cognitive function, and alcohol and substance use. (A)** The geometry of behavioral PC 1 (black, left circle) reflects the difference in group-average behavioral variables (standardized behavioral data, right circle) between subgroups **A** (yellow) and **B** (green). Subgroup **C** is not shown because no significant group differences are found in **(D)**. **(B)** The geometry of behavioral PC 2. **(C)** The geometry of behavioral PC 3. **(D)** Comparison of individual PC 1 scores between subgroups identified using neural state-trait measures (**Fig 4**). Two-sample two-sided *t* tests were performed between subgroups for each behavioral PC. $p_{BON}$: Bonferroni corrected *p*-values. **(E)** Multiple linear regression model of 3 neural PC 1 with 2 covariates (age and sex) showed that the neural PC 1 was associated with the behavioral PC 1 (Partial $R^2 = 0.023$, $\beta_1 = 0.26$, *SE* = 0.09, *t* = 2.8, *p* = 0.006), where multiple $R^2 = 0.041$, adjusted $R^2 = 0.026$, *F*(5,331) = 2.814, *p*-value = 0.017 for the full model. **(F–H)** Reproducibility analysis of the prediction of individual behavioral PC scores from neural PCs. In each permutation, PCA was performed for the neural and behavioral data from subjects in a random half of the entire sample (*N* = 168). Parameters of multiple linear regression models with three neural PC 1 with 2 covariates (age and sex) were estimated to evaluate the predictability of each behavioral PC. $p_{BON}$: Bonferroni corrected p-values from *F*-tests. The data used to generate the results can be found in **S2** and **S3 Data**. PCA, principal component analysis.

(**Fig 6E**). When comparing the individual scores of behavioral PC 1 between sex, we found no relationship. We did not observe any behavioral relevance of neural state-trait dynamics in identifying subgroup **C** (*n* = 47). In addition, behavioral PC 3 showed a strong sex effect ($p_{BON} < 0.005$).

Next, we studied if individual scores on the behavioral PC 1 are associated with individual scores on the 3 neural PCs using the multiple linear regression model (behavioral PC 1 ~ neural PC 1 + neural PC 2 + neural PC 3 + age + sex). The neural PC 1 was associated with the behavioral PC 1 (partial $R^2 = 0.023$, $\beta_1 = 0.26$, *SE* = 0.09, *t* = 2.8, *p* = 0.005), where the multiple $R^2 = 0.041$, adjusted $R^2 = 0.026$, *F*(5, 331) = 2.814 and *p*-value = 0.017 for the full model for predicting the behavioral PC 1 (**Fig 6E**). The neural PCs 2 and 3 and age did not show any association. Sex exhibited a weak association with the behavioral PC 1 (partial $R^2 = 0.016$, $\beta_1 = -1.44$, *SE* = 0.61, *t* = -2.34, *p* = 0.02).

### Reproducibility and cross-validation of low-dimensional neuro-behavioral association

To further evaluate the reproducibility of the neuro-behavioral association in our low-dimensional space found in **Fig 6E**, we first performed the same multiple linear regression approach on a split data (random $N = 168$) across 1,000 permutations. Null data were generated by shuffling individual subjects in each behavioral PC data (**S15 Fig**). For predicting behavioral PC 1, the resulting partial $R^2$ values were strongly reproducible across permutations (partial $R^2 = 0.025\pm0.017$ for neural PC1, $p_{BON}<e-10$, $F$-test; overall $R^2 = 0.056\pm0.024$), as shown in **Fig 6F**. Similar to the analysis using the entire data, sex was also a reproducible predictor of behavioral PC 1 (partial $R^2 = 0.02\pm0.016$, $p_{BON}<e-10$, $F$-test). Secondly, we used the multiple linear regression model trained from each split 1 data for predicting individual behavioral scores in the corresponding split 2 data across permutations (**S16 Fig**). While the overall prediction performance was relatively low, it was highly reproducible and significantly different from null data analysis ($R^2 = 0.011\pm0.013$, $p < e-10$, $F$-test).

Next, we repeated the same analyses for predicting behavioral PCs 2 and 3. Neural PCs 1 and 2 showed reproducible association with behavioral PC 2 that highlights emotion, personality, and psychiatric life functions, whereas age and sex showed larger predictive power (**Fig 6G**). For predicting behavioral PC 2, the estimated partial $R^2$ was $0.008\pm0.01$ for neural PC2, $0.018\pm0.014$ for age, and $0.013\pm0.019$ for sex (**Fig 6G**). On the other hand, neural PC 2 was predictive of behavioral PC 3 highlighting a strong association with alcohol consumption habits and sex differences (**Fig 6H**). For predicting behavioral PC 3, the estimated partial $R^2$ was $0.016\pm0.014$ for neural PC2 and $0.059\pm0.036$ for sex, whereas the overall $R^2$ for the full model was $0.098\pm0.041$ (**Fig 6H**).

### Impact of CAP III on the principal neuro-behavioral relationships

It remains unclear whether and how the presence of CAP III impacts the temporal CAP profiles of other CAPs and how it relates to individual differences in behavior. To address these, we studied the relationship of CAP III to the 3 neural PCs (**Fig 4**) and the first behavioral PC (**Fig 5**). Specifically, to quantify the probability of CAP III occurrence, we compared the probability to have 5 CAPs involving CAP III and the probability to have 4 CAPs without involving CAP III. We found that subgroup **C** had a high probability of CAP III occurrence, when compared to other subgroups (**Fig 7A**). Individuals that have a high probability of CAP III occurrence present low scores of neural PC 1 ($r = -0.26$, $p<0.001$) and high scores of neural PC 2 ($r = 0.24$, $p<0.001$; **Fig 7B and 7C**). There was no relationship to individual scores of neural PC 3 (**Fig 7D**). There was a weak negative correlation between the probability of CAP III occurrence and individual scores of behavioral PC 1 ($r = -0.18$, $p<0.005$; **Fig 7E**). We found no correlation between the probability of CAP III occurrence and behavioral PCs 2 and 3. These results together indicate the association of spatiotemporal properties of CAP III with the neural PCs and the behavioral PC 1.

### Impact of motion

To evaluate the impact of motion, the mean frame displacement (FD) was computed across time frames for each subject. The estimated mean FD was $0.16\pm0.58$ across subjects ($N = 337$). Note that we scrubbed time frames with excessive motion (FD $> 0.5mm$) when estimating CAPs (**S1 Fig**). The average number of scrubbed time frames across subjects was $76\pm192.6$ (counts), which are $1.73\pm4.38\%$ of the total number of time frames (4,400/subject before removing dummy scans) in each subject. More than 5% of total time frames were scrubbed in

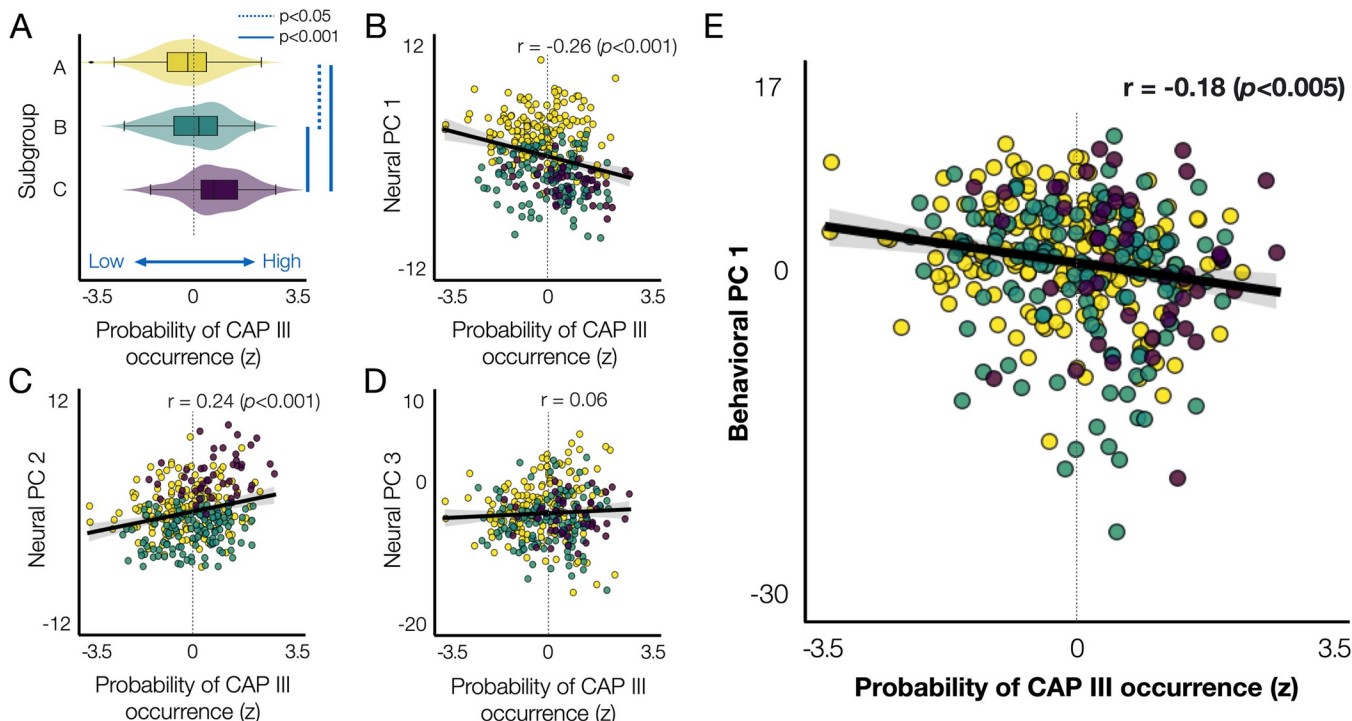

**Fig 7. The probability of CAP III occurrence is associated with the neural and behavioral PCs. (A)** The probability of CAP III occurrence ($x$-axis) for each individual, which can be interpreted as an individual's preference to have CAP III, was evaluated by the difference in the occurrence of 4 CAPs versus 5 CAPs, as described in **Fig 1B**. For each subject, we computed the number of permutations (occurrence out of 1,000 permutations) when 4 CAPs were estimated and the number of permutations for the same subject to be involved when 5 CAPs were estimated. Then, for each subject, we compared the difference in the occurrence ($\Delta$ Occurrence = Occurrence (k = 5)–Occurrence (k = 4)) from each split. Then, for each individual, the $\Delta$ Occurrence was averaged over 2 splits. Finally, the within-subject average $\Delta$ Occurrence was normalized across subjects to $z$-scores. Individuals were color-coded by subgroups defined using the hierarchical clustering of 30 neural features (**Fig 4**). **(B–D)** Scatter plots of individuals' preference to have CAP III with respect to the individual scores on the neural PC 1 **(B)**, neural PC 2 **(C)**, neural PC 3 **(D)**, and behavioral PC 1 **(E)**. The data used to generate the results can be found in **S4 Data**. CAP, co-activation pattern.

28/337 subjects (8.3%). We also measured the duration of motion (the number of consecutive time-frames with excessive motion) to assess whether there were long time-segments of motion, which might impact the estimation of temporal CAP profiles, especially analyses of CAP state transitions and dwell time. The length of motion-related continuous time-frames was 1.34±0.38 on average across subjects. Repeating the analyses excluding these 28 subjects did not change the results, including the PCA of neural measures and the low-dimensional neuro-behavioral relationships (**S17 Fig**).

In summary, we tested the hypothesis that there is a reproducible CAP feature set that reflects both state and trait brain dynamics and that this combined feature set relates to individual phenotypes across multiple behavioral domains. Specifically, **behavioral PC 1** highlights individual life function outcomes associated with cognition, emotion regulation, alcohol use, and substance use. Individuals with high behavioral PC 1 scores are found to (i) spend longer time at CAP I than at CAP II; (ii) have higher between-subject variance and lower within-subject variance at CAP I than at CAP II; (iii) show high global persistence (longer dwell time) in all CAPs; and (iv) lower chance to have CAP III. Such neuro-behavioral patterns were (v) associated with sex differences. **Behavioral PC 2** highlights emotion regulation, personality and psychiatric life functions. Individuals with high behavioral PC 2 scores are found to (i) show longer dwell time in all CAPs; (ii) spend longer time at CAP I than at CAP II; and (iii) be associated with age. **Behavioral PC 3** highlights alcohol, nicotine and substance use.

Individuals with high behavioral PC 3 scores are found to (i) have longer dwell time in all CAPs; (ii) higher chance to have CAP III; and (iii) be associated with sex differences.

## Discussion

This study provides evidence to highlight the importance of quantifying both within-subject and between-subject variance components of brain dynamics and their link to individual differences in functional behavioral outcomes. Here, we show that the dynamics of rs-fMRI can be quantified via CAP analyses and reveal reproducible neural features that can maximize effects of state variance, trait variance, and test-retest reliability. We found that the human brain manifests a highly reproducible low-dimensional set of features that index brain-wide co-activation patterns. The neural feature reduction captures a general motif of individual variation, such that individuals occupy this low-dimensional state-trait neural space differently, which in turn predicts life and behavioral outcomes. State variance indexed by day-to-day variability in co-activation dynamics are loaded on this low-dimensional motif of individual variation, therefore reflecting neuro-phenotypes.

We identified 3 CAPs representing recurrent snapshots of mixed resting-state networks in healthy young adults, which exhibit distinct spatiotemporal profiles that are reproducible at the single-subject level. In turn, 3 subgroups of individuals were identified using hierarchical clustering of temporal CAP profiles, which mapped onto distinct aspects of CAP dynamics capturing both state (i.e., within person) and trait (i.e., between person) variance components. We found that the principal variations of neural state and trait CAP features co-vary with the principal variations of behavioral phenotypes. Put differently, we identified specific properties of rs-fMRI dynamics that mapped onto a person's life outcome profile. Critically, person-specific probability of occupying a given CAP was highly reproducible and associated with the neural and behavioral features. Collectively, these results show that a reproducible pattern of neural dynamics can capture both within-person and between-person variance that quantitatively map onto distinct functional outcomes across individuals.

### Identifying reproducible neural dynamics profiles in humans

In this study ($n = 337$, **Fig 1E**), we identified 3 reproducible CAPs. These CAPs captured spatial patterns similar to the analysis results of zero-lag standing waves and time-lag traveling waves of rs-fMRI BOLD fluctuations previously identified by Bolt and colleagues, using complex PCA and a variety of latent dimension-reduction methods for the HCP data set ($n = 50$) [52]. The spatial correspondence between the 3 patterns identified by Bolt and colleagues and the CAPs discovered in our study aids in the interpretation of our results. Specifically, the spatial topography of CAPs I+/I- may be linked to task-positive/task-negative dynamics of BOLD signals, while CAPs II+/II- may be associated with global signal fluctuations [52]. However, similar to most early studies on CAPs in rs-fMRI [33], Bolt and colleagues employed a sparse time point sampling strategy (15%) based on high-amplitude signals of time courses in predefined regions, along with an arbitrary choice of two-cluster solution [52]. The sparse time point sampling is based on a hypothesis that patterns of functional connectivity arise from discrete neural events [6], often driven by high-amplitude co-fluctuations in cortical activity [53]. These studies demonstrated the spatial correspondence between estimated CAPs and widely studied resting-state functional connectivity patterns, such as the default mode network [6,33,54].

Nevertheless, no study to our knowledge has investigated the joint properties of within and between-subject variation of CAPs patterns across the entire BOLD signal range. Additionally, no study has examined the impact of considering the full BOLD signal range on the

relationship between CAP properties and behavior [36–42]. Here, we present an analytic approach that optimizes within-subject variance, between-subject variance, and test-retest reliability of identified CAPs using the entire BOLD signal range. Critically, we demonstrate reproducible spatiotemporal CAP features for each subject (**Figs 2**, **3**, **S9** and **S10**). In turn, we show an association between the principal variations of CAP neuro-phenotypes and the principal variation of behavioral phenotypes (**Fig 5**).

It was beyond the scope of the present study to assess the impact of temporal sampling of BOLD signals on CAP analysis. However, previous research have suggested that focusing only on particular time points, such as those during events with high BOLD signal amplitude or strong signal correlation with a seed region, while disregarding the remaining of the data (e.g., event-absent), may potentially result in misleading conclusions [44]. Iraji and colleagues [44] noted that event-absent time points could capture a unique and robust relationship between the default mode functional network connectivity and schizophrenia symptoms. Therefore, future investigations addressing this specific question will be helpful for determining the extent to which transient brain co-activation patterns can capture diverse characteristics of individual brain dynamics. For instance, CAP derived measures can be quantified from data with various temporal sampling strategies, such as varying the proportion of analyzed time points randomly or based on specific signal criteria. In addition, the impacts of temporal resolution of fMRI BOLD acquisition (e.g., time of repetition; TR) should be given further consideration.

A recent work suggests a method to detect individual CAPs at the subject-level by maximizing individual identifiability [55]. Using the densely sampled Midnight Scan Club data set [56], they identified four CAPs at both the group-level and single-subject level, with 2 CAP pairs exhibiting opposing spatial patterns, similar to the findings in this work (**Fig 1**). However, their study did not address how individualized CAPs and their temporal profiles can be assessed in relation to behavior, focusing on an identical number of clusters at both levels. In contrast, we focused on identifying reproducible spatial patterns of neural co-activation (CAPs) at the group level and quantifying a reproducible feature set of temporal profiles of the CAPs at the single-subject level, therefore providing a unified framework to evaluate individual differences using a joint analysis of state and trait variations of neural co-activation. Similarly, several brain states were estimated at the group level using another dynamic functional connectivity approach in [24], whereas they could identify the time-points when the state was active, allowing the estimation of FO for each state and for each subject.

Collectively, these results highlight that state-trait CAP dynamics are reproducible at the single-subject level across permutations and splits (**Figs 3** and **S9**). For context, the statistics reported here (**Fig 3C**) demonstrate higher reproducibility than the meta-analytic estimate for group-level reproducibility of area-to-area functional connectivity matrices [51]. Still, the observed ICC values are fairly low, reflecting a notable variance between days. This indicates that while trait-like features are most dominant factors loaded on the general motif of individual variance, the observed state variance (between days) also contributes to this general motif. In this context, the fairly low range of ICC reflects the notable amount of within-subject variance at the time-scale of days, and supports our framework of state-trait components that together identify neural phenotypes. Similarly, Yang and colleagues found that CAPs identified at individual level were unstable over time across the 10 scans (~ 30 min/scan) except a few subjects, and subject-specific CAPs became more reliable and individual specific when integrating data with longer duration [55]. Therefore, we argue that such temporal fluctuations within subjects can bring critical insights into individual-specific brain organizations. Reducing the number of neural features into a reproducible set of CAPs may enable a more robust and reproducible mapping between neural features and behavior. In other words, we hypothesize that further optimization of reproducible data-reduced neural features presents a critical

step toward mapping rs-fMRI signals to healthy and clinically relevant behavioral variation and obtaining robust neuro-behavioral models.

## Quantifying joint state and trait variance components of neural dynamics

The three-axes representation of spatiotemporal CAP dynamics, illustrated in **Fig 3E**, highlights an approach to consider temporal CAP characteristics that can inform feature selection. Put differently, we show that by projecting CAP measures derived within each subject into a trait variance space, it is possible to visualize how CAP properties that vary within a person (state) also vary between people (trait).

For example, we found that CAP II exhibits the highest relative between-subject variation (i.e., trait) across all measures presented here. Conversely, CAP III exhibits lower between-subject variance but higher within-subject variance than CAP II. This suggests that, although there is less individual variation in CAP III overall, any given person may exhibit marked variation in this pattern between days. These observations were highly reproducible and were generally agreed with the variance explained by the 3 patterns reported in [52]. This raises the question of whether the joint consideration of both state and trait metrics can reveal key properties of neural features that, in turn, can inform their mapping to behavior. For example, one would expect that a neural feature that varies markedly between individuals but shows little within-subject variance may serve as a reliable neural marker for tracking longitudinal behavioral changes (e.g., neurodevelopmental changes or rapid mood swings observed in certain psychiatric populations, which may not occur in healthy populations). In contrast, neural features that maximize within-subject variation, while still exhibiting notable trait variance, may be better at detecting neuro-behavioral relationships expected to undergo substantial changes over time.

Indeed, using both state and trait variance components of identified CAPs revealed 3 subgroups of healthy subjects. This finding aligns with the notion that using neural features with distinct patterns of state variances can provide vital information about individual differences (**Fig 4**). The objective of this clustering was not to categorize individual subjects. Rather, we aimed to test whether there exists a set of neural features commonly observed across a number of healthy subjects, exhibiting reproducible neural co-activation properties that can be related to behavioral phenotypes. We first found that the 3 subgroups ($n$ = 163, 127, and 47 for each group) could be projected into a data-reduced PCA model. Neural PC 1 is characterized by distinct patterns of FO and DT measures between CAPs I/III versus CAP II (CAP-specific), neural PC 2 represents the general persistence of all CAP states (general), and neural PC 3 represents day-to-day variations within individuals (**Fig 4D–4F**). This additional level of neural feature reduction captured a general motif of how individuals vary in terms of complex temporal patterns of neural co-activation.

While this study suggests the importance of taking state variations into account when studying neural basis of individual differences, we did not directly compare the performance of neuro-behavioral mapping when using only state features, only trait features and both state and trait features. Further studies with comprehensive experiments on state manipulations such as pharmacological neuroimaging or transcranial magnetic stimulation (TMS) can help to understand how state and trait neural features change in response to such manipulations and whether the state and trait neural features reflect unique or additive information about individual differences.

## Linking neural patterns of co-activation to behavioral and life functioning

One of the key goals in human neuroimaging is to identify features that relate to human function. More specifically, do signals derived from fMRI carry information that can be related to

positive or negative life functional outcomes in adults? Prior work tested this hypothesis using multi-variation canonical correlation approaches (CCAs) [22]. While these initial findings were compelling, it is not widely appreciated that CCA models that use many neuroimaging features are prone to overfitting. To address this issue, we investigated whether the reduced and reproducible neural feature set, identified by the joint state and trait variance components of neural dynamics, can explain variation in functional behavioral outcomes in a sample of adults representative of the general population. Here, we computed a PCA model on 262 behavioral features from the HCP sample, which revealed a solution with $n = 27$ PCs that passed permutation testing. However, we found that the first behavioral PC captured >11% of all behavioral variance and it was highly reproducible (between-split correlation of behavioral PC 1 loadings was r > 0.9; **Fig 5C**). Therefore, we examined the relationship between the first CAP-derived neural PC (**Fig 4**) and the first behavioral PC, which revealed that individuals with higher neural PC 1 scores (subgroup **A**, **Fig 4F**) also have higher behavioral PC 1 scores (**Fig 6D and 6E**).

These results suggest that individuals who preferentially occupy CAP I and exhibit strong state persistence also demonstrate higher cognitive and affective functional outcomes (**Figs 4** and **5D**). In contrast, individuals who predominantly occupy CAP II for extended periods tend to exhibit relatively lower cognitive scores, along with higher levels of alcohol and substance use. This aligns with the notion that general brain-wide patterns of co-activation in fMRI signal are associated with an individual's level of functioning. Of note, CAP II exhibited the highest relative between-subject variation across all measures (**Fig 5D**). Furthermore, CAP II showed a spatial motif that appeared to be "global." This is consistent with prior findings showing that a global rs-fMRI signal topography, which contained a major contribution of the fronto-parietal control network, was associated with positive and negative life outcomes and psychological function [57]. Interestingly, we found that observing CAP III might be related to the composition of the studied sample. In other words, there is a group of people with high occurrence of CAP III (subgroup **C**), which if sampled in the reported permutation testing will yield a 3-CAP solution (I, II, and III). A higher probability of CAP III presence across individuals was associated with lower behavioral PC 1 scores, indicating poor functional life outcome (**Figs 1**, **2** and **6**). More specifically, individuals with high probability of CAP III neural signal pattern exhibit relatively lower cognitive function, higher alcohol use, and higher substance use.

Our results converge with several findings using other dynamic functional connectivity approaches. Vidaurre and colleagues used hidden Markov model to identify 12 brain states using the HCP-YA rs-fMRI data sets and derived 2 metastates, each being a set of brain states that are more likely to transit between each other [24]. They found that the FO of brain states and their associated metastates are subject-specific and behaviorally relevant, highlighting several well-being, intelligence and personality traits [24]. It would be interesting to evaluate, using the same subjects studied in this work, if brain activity during cognitive tasks exhibit a similar set of CAP features and low-dimensional state-trait variances to those estimated using rs-fMRI and how they are related to such high-order metastates [24] and the low-dimensional global brain activity found across multiple cognitive tasks [26].

This strongly supports the idea that reproducible functional co-activation patterns in the human brain can map onto behavioral outcomes that have implications for mental health. Here, we found this pattern by considering only the first PCs of the neural and behavioral feature spaces. It remains unknown whether further feature optimization of CAP dynamics would reveal stronger effects in relation to more severe mental health symptoms, which can be detected in clinical samples. In fact, spatial and temporal organization of CAPs has been linked to psychiatric symptoms in previous work [37–42]. However, it is unknown if the neural

features derived from CAPs that are reproducible in the healthy general adult population are also predictive of severe psychiatric symptoms. In other words, it is possible that there are CAPs (and associated state-trait variance components we quantified) that are only detectable in individuals who experience a certain level of symptom severity. In this context, it is vital to consider the likelihood and the timescale on which state neural measures are defined—namely how likely is a state to be present in a person and how long does it last to be relevant for behavior. Relatedly, it is key to consider how much between-person variation there has to be in a given CAP state pattern to reveal individual symptom variation across a clinical sample—thus making it a trait-like neural marker of psychiatric symptoms. The results of this study highlight how critical it might be to parse transient (state) or persisting (trait) CAP properties when it comes to clinical applications.

In other words, mental health symptoms can be considered to vary between people (i.e., as a trait) or vary within a person (i.e., as a state), which can be quantified separately. Trait anxiety, for example, is the tendency of a person to experience anxious affect across a broad range of contexts and for extended periods of time. In contrast, state anxiety is clinically defined anxiety occurring in the present moment [58,59]. The current findings suggest that the probability of exhibiting high anxiety in general and the likelihood of being anxious at any given moment may be linked to the same underlying neural co-activation pattern occurrence. We posit that this may be a general phenomenon that can be extended to other mental health outcomes. Therefore, it would be valuable in future work to study the combined contributions of state and trait neural features in predicting the severity and likelihood of occurrence for a mental health outcome [60].

Our results in the reproducible neuro-behavioral associations highlighted the neural relevance of behavioral PCs 1, 2, and 3. Individuals with higher cognition (behavioral PC 1) spent longer time at CAP I than at CAP II. The behavioral PC 2 showed an association with neural PCs 1 and 2, as well as age and sex. In relation to resting state functional connectivity, while CAPs I and II show differences in the regions of language, primary visual, and cingulo-opercular networks, the overall patterns of these CAP states involved opposing patterns of brain activity in the default mode and frontoparietal networks versus the regions belonging to the secondary visual, somatomotor, and dorsal attention networks [61]. This suggests a relation to the cortical hierarchy that spans from unimodal sensorimotor cortices to transmodal association cortices and a potential role in functional connectivity development during childhood through adolescence [62].

Finally, an important consideration here is that we did not evaluate the impact of sample size on the estimation of CAPs and their properties. It is unknown if the low-dimensional feature set that captures a general motif of individual variation found in this study is particular for the samples in the HCP S1200 data set or generalizable to larger and more diverse samples of the general population. In data sets with dense sampling over days, months, or years, we may find another dimension of within-subject (state) variation that contributes to the general motifs of individual differences. It is possible that with a smaller sample size or different composition of the sample, there might be a reduced chance of observing a specific CAP (e.g., CAP III) or even detect new CAPs. This could occur because a particular CAP may be rare, especially when it relates to a neural pattern that is uncommon in the general population, which may be the case for neuropsychiatric or neurological symptoms. Another important aspect to consider is the extension of this work to pediatric and adolescent samples, given that there may be a substantially different configuration of CAPs as the human brain develops. Besides, in the present study, we focused on examining the patterns of CAPs in healthy young participants with no family relation. Using other dynamic functional connectivity approaches (e.g., hidden Markov model), it has been suggested that the

FO of brain states are subject-specific and highly heritable [24]. Exploring the full HCP S1200 data set in future work may help to address these questions and whether these CAPs and related neural measures are heritable and more similar between monozygotic or dizygotic twins.

## Biological interpretation of CAPs

In general, CAPs derived from BOLD fMRI signals can be interpreted as patterns of how different brain regions synchronize and coordinate their activity over time. The 3 CAPs identified in this study represent brain-wide co-activation motifs that can be reproducibly observed in the general populations at the given temporal and spatial resolution. Although these 3 CAPs, with positive and negative signs, may not encompass all possible brain states, it is notable that our result converges on such a reproducible low number. While the CAP analysis is dependent on the number of samples (subjects) and time-points, in this study, this limitation was balanced by determining the number of CAPs by what was found to be reproducible. However, we acknowledge that the true number of CAPs in the human brain may exceed what is established in this analysis. Future research using larger datasets could reveal additional reproducible brain states, including rare but consistent CAPs. Potential factors contributing to individual variability during rs-fMRI scans, such as extreme anxiety, heightened emotion, ongoing cognition or physical pain during resting-state scans, might impact CAP estimation. Investigating this variability in detail could help identify potential motifs that were previously considered rare or non-reproducible in smaller samples, but may become evident in larger data sets, such as the full samples from HCP-YA S1200 dataset or UK Biobank [63].

## Conclusions

Understanding how the brain generates co-activated patterns of neural activity over time is critical to derive reproducible brain-wide patterns of neural dynamics that occur in humans. Here, we advance this goal by quantifying state (within-subject) and trait (between-subject) variance components of neural co-activations. We do so by leveraging rich spatial-temporal information embedded in the entire range of rs-fMRI BOLD signals, which reveals 3 CAPs that reflect brain-wide motifs of time-varying neural activity. Critically, we demonstrate a reproducible estimation of spatial CAP features at the group level and the temporal characteristics of CAP states at the single-subject level. We found that distinct parameters of CAP temporal characteristics, such as occupancy and persistence, can be studied together and represented as either state or trait features. In turn, we show that a low-dimensional neural feature space captures both state and trait variation in CAP parameters, which in turn exhibit behaviorally relevant characteristics. Specifically, people who showed longer time spent in a given CAP, longer persistent periods within a CAP, as well as higher variation in transitioning between all CAPs, also showed higher cognitive function, improved emotion regulation, and lower alcohol and substance use. Critically, person-specific probability of occupying a particular CAP was highly reproducible and associated with both neural and behavioral features. This highlights the importance of studying CAP-derived measures as a neural marker that may be altered as a function of mental health symptoms and may change developmentally. Collectively, these results show a reproducible pattern of neural co-activation dynamics in humans, which capture both within- and between-subject variance that in turn maps onto functional life outcomes across people.

## Materials and methods

### Human Connectome Project (HCP) data set [45]

Participants were recruited from Washington University (St. Louis, Missouri, United States of America) and the surrounding area. We selected participants from the S1200 release of the HCP who had no family relations, resulting in a total of 337 participants included in our analyses. The data set contains resting-state fMRI data from 180 females and 157 males, with age range 22 to 37 (mean age = 28.6, SD = 3.7), 90% right-handed. Whole-brain echo-planar imaging data were collected with a 32 channel head coil on a modified 3T Siemens Skyra (Connectome Skyra) at WashU with time to repetition (TR) = 720 ms, time to echo (TE) = $33.1 ms$, flip angle = 52, bandwidth = 2,290 Hz/pixel, in-plane field of view (FOV) = $208 \times 180 mm$, 72 slices, and 2.0 mm isotropic voxels, with a multi-band acceleration factor of 8. rs-fMRI blood-oxygen-level-dependent (BOLD) images were collected over 2 days. On each day, 2 runs (14.5 min/run) of rs-fMRI were collected with opposite phase encoding directions (L/R and R/L). Subjects were instructed to keep eyes open with fixation on a cross-hair. Task-based imaging data were also collected, but not used in the present study. Structural MRIs were collected using the following parameters: T1-weighted (0.7 mm isotropic resolution, TR = 2,400 ms, TE = 2.14 ms, flip angle = 8, in-plane field of view = $224 \times 224$) and T2-weighted (0.7 mm isotropic resolution, TR = 3,200 ms, TE = 565 ms, variable flip angle, in-plane field of view = $224 \times 224$). Find additional details about the data set in [64].

### Data preprocessing

We preprocessed rs-fMRI using the following steps corresponding to the steps advanced by the HCP consortium: (i) the "minimal preprocessing" pipeline outlined by [65], which involves intensity normalization, phase-encoding direction unwarping, motion correction, and spatial normalization to a standard template MSMAll [66], Angular Deviation Penalty (ADP) version; (ii) high-pass filtering (0.009 Hz); (iii) ICA-FIX for artifact removal [67]. Next, the "minimally preprocessed" rs-fMRI in each run was represented in the Connectivity Informatics Technology Initiative (CIFTI) file format that combines surface-based data representation for cortex and volume-based data for subcortex gray matter locations (i.e., "grayordinates"). Additional analyses were performed with Workbench v1.2.3 and Matlab 2014b (The Mathworks), using Quantitative Neuroimaging Environment and Toolbox (QuNex) [23,47].

Previous studies focusing on CAP analysis showed consistent CAP properties across the voxel and region levels [37,68]. To analyze CAPs at a low-dimension space and to reduce the computational burden of CAP analysis that treats every 3D time frame in the clustering process, we applied the Cole-Anticevic Brain Network Parcellation (CAB-NP) parcellation [46]. The CAB-NP parcellation is comprised of (i) 180 bilateral cortical parcels (a total of 360 across both left and right hemispheres), consistent with the Human Connectome Project's Multi-Modal Parcellation (MMP1.0) [66]; and (ii) 358 subcortical parcels defined using resting-state functional BOLD covariation with the cortical network solution [46]. To remove any potential artifact at the onset/offset of each run, the first 100 frames were removed from every rs-fMRI run for each subject. To normalize rs-fMRI data in each run, the mean of each run was removed from each time series. Subsequently, rs-fMRI BOLD runs were concatenated in order of acquisition (rs-fMRI runs 2-1-4-3, R/L first, then L/R), resulting in a $4,000 \times 718$ array of rs-fMRI data for each subject.

## CAP analysis

We identified moment-to-moment changes in the whole brain rs-fMRI BOLD signals at each time point and quantified the spatial patterns of co-activation (CAPs) across individuals, as well as individual variations in CAP temporal organization [33]. The analytic framework proposed in this study is described in **S1 Fig** and implemented using Python 3.6.15 using the Yale High Performance Computing resources. In each permutation, $N = 337$ subjects are randomly split into 2 equal-sized groups $n = 168$, nonoverlapping subjects. We used the shuffled split-half resampling strategy for several reasons. First, applying K-means clustering once to the concatenated time-series of the entire sample results in single values of neural measures per subject. Using such a simple approach, the individual reliability of CAP measures from the concatenated time-series across various sample populations could not be quantified. Our approach allowed for the estimation of statistical reproducibility of neural measures for each subject, when a subject was included in different sample populations, therefore reducing potential sampling bias. In addition, we could use the same split-half resampling scheme for the cross-validation of neuro-behavioral association analysis, by training a neuro-behavioral model from split 1 data and testing the model on the remaining split 2 data.

Within each split, a $4,000 \times 718$ array of preprocessed rs-fMRI data are temporally concatenated across subjects. The following steps were performed for each split data using *scikit-learn 1.3.2* with Python. (i) Time-frames with excessive motions (Frame displacement > $0.5mm$) are scrubbed [23,69,70]. The HCP minimal preprocessing pipeline included motion correction [65]; therefore, we avoided using a too conservative threshold for motion scrubbing and retained as many potentially useful frames as possible in our analysis. (ii) The remaining time-frames are clustered based on spatial similarity using the K-means clustering algorithm with Lloyd's algorithm, varying the number of clusters ($k$) from 2 to 15. (iii) An optimal number of clusters $\hat{k}$ is estimated. (iv) Using the K-means solutions with the optimal number $\hat{k}$, CAPs are defined as the cluster centroids, by parcel-wise averaging of the time-frames within each cluster.

## Number of clusters

While the estimation of the number of clusters $k$ is critical in CAP analysis, the field is lacking a consensus on the optimal criterion to determine it [71]. Earlier works used predefined arbitrary numbers ranging from 6 to 30 and reported that CAPs estimated using a small $k$ are consistently found in results using a larger $k$ [7,37,72]. It has been suggested that a large $k$ may reduce cluster stability, for example, when a small number of time-frames are allocated to a cluster due to a short rs-fMRI acquisition duration [73]. Yang and collegaues calculated the silhouette scores for the clustering results varying $k$ from 2 to 21 in both group and individual level analyses and chose $k = 4$ as a trade-off to ensure the reliable estimation of spatiotemporal dynamics [55].

To find an optimal number, we used an approach that considers a trade-off between the number of clusters and within-cluster similarity (e.g., silhouette criteria [74]). We observed that the silhouette score was monotonically decreasing with the increase of $k$ (**S2 Fig**), in agreement with Yang and colleagues [37]. In our study, to determine an optimal number of clusters, we varied the number of clusters ($k$) from 2 to 15. For each $k$, the K-means clustering was initialized using the k-means++ algorithm, by selecting randomly generated centroids using sampling based on an empirical probability distribution of the points' contribution to the overall inertia. Inertia was defined as the sum of squared distances of samples to their closest cluster center. The maximum iteration for a single run was set to 1,000 to avoid that the algorithm stops before fully converging. Silhouette coefficient is estimated for the K-means solution

using each $k$. Finally, an optimal number $\hat{k}$ is determined by applying the elbow method for the estimated Silhouette scores. The elbow point was defined when we observe a significant change in the rate of decrease of the Silhouette score as $k$ increases, using the KneeLocator class in Python's kneed package.

## Basis CAP generation

The occurrence rate (%) of the $\hat{k} = a$ solution was calculated by the number of permutations resulting in $a$ clusters divided by the total number of permutations (1,000). Co-occurrence rate (%) of the $\hat{k} = a$ solution in both splits was determined by the number of permutations resulting in the same number of clusters divided by the total number of permutations. See Part (1) of the procedure diagram in **S4 Fig**.

A set of basis CAPs can be obtained using the agglomerative hierarchical clustering of the CAPs estimated from the permutations resulting in the same number of clusters ($\hat{k} = a$), as follows. For each split, let's first denote that $h_{MAX}/1,000$ permutations resulted in $\hat{k} = a$ solution. (1) We collect the $\hat{k}$ CAPs ($P \times \hat{k}$) and concatenate them across $h_{MAX}$ permutations to produce a ($P \times \hat{k} h_{MAX}$) array, where $P$ is the number of parcels. (2) Agglomerative hierarchical clustering is applied to this array to identify $\hat{k}$ clusters based on spatial similarity. (3) In each cluster, co-activation values in each parcel are averaged across CAPs assigned to the same cluster, generating an average (basis) CAP. (4) The values in each parcel of the basis CAP are normalized to Z-scores, using the mean and standard deviation across the parcels in the whole brain. Steps (3) and (4) are repeated for all $\hat{k}$ clusters, resulting in $\hat{k}$ basis CAPs. See Part (2) of the diagram in **S2 Fig**.

## Individual preference for a specific CAP

The probability of CAP occurrence, which can be interpreted as an individual's probabilistic preference for a specific CAP, was quantified by examining the number of permutations that resulted in a specific solution $k$. To do this, we compared the probability to have $k$ CAPs involving the CAP of interest and the probability to have $k-1$ CAPs without involving the CAP of interest, assuming a reproducible estimation of spatial topography of $k$ CAPs across permutations, similar to the approaches comparing full and reduced models. Specifically, across 1,000 split-half permutations, a subject is involved in split 1 data for $p_1$ permutations and in split 2 data for $p_2 = 1,000 - p_1$. Then, when only considering split 1 data from these $p_1$ permutations, we can compute the number of permutations that resulted in $k$ and the number of permutations that resulted in $k-1$. In each split, for each subject, we compute the difference (occurrence of $k$ CAPs) minus (occurrence of $k-1$ CAPs) to quantify an individual's preference for a specific CAP. To associate these with behavioral variables (normalized), we normalized the individual's probabilistic preference for a specific CAP using Z-transformation across subjects.

## Quantification of neural measures and the impact of motion

Fractional occupancy (($FO(s, i)$)), is defined as the total number of time frames ($\#TR$) that a subject $s$ spends in CAP state $i$ per day, normalized by the total number of time frames spent in any CAP state by subject $s$ per day. Dwell time (($DT(s, i, c)$)) is defined as the number of time frames ($\#TR$) of a time-consecutive segment $c$ occupying the same CAP state $i$ within a subject $s$ per day. Within-subject mean and variance of DT, Mean $DT(s, i)$) and Var $DT(s, i)$, were computed as the mean and standard deviation of estimated values of DT from all time-

consecutive segments occupying a CAP $i$ within a subject $s$ per day. These CAP measures were estimated for each split data per permutation and for each day separately. Note that time frames with excessive motion (FD > 0.5 mm) were scrubbed before computing these CAP measures [47,69]. Motion scrubbing is helpful in minimizing the deleterious impacts of motion on the analysis of CAPs and CAP-derived measures, but it can potentially affect the quantification of these CAP measures [75]. While CAP states during motion-related time frames remain unknown because scrubbing is performed before the CAP identification, the time frames before and after the motion-related time frames would be combined into a single consecutive segment if they occupy the same CAP state. For example, if a consistent CAP state (e.g., [I,I,I], true $DT$ = 3) is interrupted by a high motion volume $M$ (e.g., [I,M,I]), the estimated DT would be shorter than the true value ([I,I], estimated $DT$ = 2). Conversely, if the CAP state during motion differs from that before and after ([I,II,I], true $DT$ = 1 for each of three segments), DT would be estimated for the combined segment after scrubbing [I,I], resulting in $DT$ = 2. To verify that our findings were not impacted by this effect, we investigated the amount of motion (number of scrubbed time frames) and the duration of motion (number of consecutive time frames with excessive motion). We identified subjects whose fMRI time-series involved relatively high motion, resulting in the scrubbing of more than 5% of total time-frames, and repeated the whole analyses excluding these subjects.

## Neural dimension reduction

To identify the principal geometry of the state-trait neural feature space, 30 neural features are estimated for each individual: 3 neural measures (FO, mean DT, and var DT) × 5 CAPs (I+, I-, II+, II-, and III) × 2 days. These neural features were collected across subjects to create a subject-by-feature matrix. Two analyses are performed on this subject-by-feature matrix. First, agglomerative hierarchical clustering was applied to the feature matrix, using *scikit-learn 1.3.2*. The ward linkage criterion with Euclidean metric was used to minimize the variance of the clusters being merged. The number of clusters was determined using a distance cut-off value of 70% of the final merge in the dendrogram. Second, PCA was applied to this subject-by-feature matrix to estimate the principal geometry of this state-trait feature space identifying subgroups.

## Behavioral data analysis

The analysis of behavioral data was implemented using the method described in [23]. We performed PCA on 262 variables across 15 behavioral domains from the HCP S1200 unrestricted and restricted behavioral data (**S14 Fig**). Behavioral variable names and the corresponding domains used in this analysis were identical to the variable names provided by the HCP data dictionary for the S1200 data release. When both age-adjusted and un-adjusted data are available, we use age-adjusted data only. To study the association between individual scores on the first behavioral PC and individual scores on the first 3 neural PCs, we use the multiple linear regression model (behavioral PC 1 ~ neural PC 1 + neural PC 2 + neural PC 3 + age + sex). The association between a neural PC and the behavioral PC 1 was assessed by calculating the partial $R^2$, regression coefficient $\beta$, and standard error ($SE$). The significance of regression coefficients was determined by computing the corresponding $t$-scores. Partial $R^2$ was defined as the coefficient of partial determination which is measured by the proportional reduction in sums of squares after a variable of interest is introduced into a model. Visualization and statistical analyses were conducted using Python 3.6.15 and R Studio v.2022.12.0.

## Ethics statement

In the collection of HCP Young Adult S1200 data, each participant provided their review and signature on the informed consent document at the start of day 1, as directed by the institutional review board (IRB) at Washington University at St. Louis, USA [45]. This study was conducted according to the principles expressed in the Declaration of Helsinki. KL, JLJ, and AA have obtained the acceptance of HCP Open Access Data Use Terms for access to all HCP data. KL further obtained the approval for access to Restricted Data generated by HCP, WU-Minn-Ox HCP. All analyses conducted in this work were approved by the IRB at Yale University (IRB number: 1111009332), Connecticut, USA. This work is 100% based on human effort, and no artificial intelligence (AI)-assisted technologies were used in the production of this article.

## Declarations

KL consults for Manifest Technologies. AA and JDM hold equity with Neumora Therapeutics (formerly BlackThorn Therapeutics), Manifest Technologies, and are co-inventors on the following patents: Anticevic A, Murray JD, Ji JL: Systems and Methods for Neuro-Behavioral Relationships in Dimensional Geometric Embedding(N-BRIDGE), PCT International Application No. PCT/US2119/022110, filed March 13, 2019 and Murray JD, Anticevic A, Martin WJ: Methods and tools for detecting, diagnosing, predicting, prognosticating, or treating a neurobehavioral phenotype in a subject, US Application No.16/149,903, filed on October 2, 664 2018, US Application for PCT International Application No.18/054, 009 filed on October 2, 2018. JLJ is an employee of Manifest Technologies, has previously worked for Neumora, and is a co-inventor on the following patent: Anticevic A, Murray JD, Ji JL: Systems and Methods for Neuro-Behavioral Relationships in Dimensional Geometric Embedding (N-BRIDGE), PCT International Application No.PCT/US2119/022110, filed March 13, 2019. CF consults for Manifest Technologies and formerly consulted for RBNC (formerly BlackThorn Therapeutics). GR consults for and holds equity in Neumora and Manifest Technologies. LP is an employee of Manifest Technologies. JHK holds equity in Biohaven Pharmaceuticals, Biohaven Pharmaceuticals Medical Sciences, Clearmind Medicine, EpiVario, Neumora Therapeutics, Tempero Bio, Terran Biosciences, Tetricus, and Spring Care. JHK consults for AE Research Foundation, Aptinyx, Biohaven Pharmaceuticals, Biogen, Bionomics, Limited (Australia), BioXcel Therapeutics, Boehringer Ingelheim International, Cerevel Therapeutics, Clearmind Medicine, Cybin IRL, Delix Therapeutics, Eisai, Enveric Biosciences, Epiodyne, EpiVario, Evidera, Freedom Biosciences, Janssen Research & Development, Jazz Pharmaceuticals, Leal Therapeutics, Neumora Therapeutics, Neurocrine Biosciences, Novartis Pharmaceuticals Corporation, Otsuka America Pharmaceutical, Perception Neuroscience, Praxis Precision Medicines, PsychoGenics, Spring Care, Sunovion Pharmaceuticals, Takeda Industries, Tempero Bio, Terran Biosciences, and Tetricus. All other co-authors declare no competing interests.

## Supporting information

**S1 Fig. Workflow of CAP analysis.**
(TIF)

**S2 Fig. Quality of the K means clustering solution in CAP analysis.** Silhouette scores were estimated across different numbers of clusters (*k*) from the K-means clustering solution from a split data. Results from 10 permutations (2 split-halves in each permutation) are shown. Optimal *k* values were estimated using the elbow method for the Silhouette scores and are

highlighted in red.
(TIF)

**S3 Fig. Occurrence of CAPs across permutations. (A)** The estimated number of CAPs ($k$) in each split across 1,000 permutations. **(B)** Occurrence rate (%) of k = 4 or k = 5 solutions in each split. **(C)** Co-occurrence rate (%) of $k$ = 4 or $k$ = 5 solutions in both splits.
(TIF)

**S4 Fig. Generation of basis CAP sets.**
(TIF)

**S5 Fig. Spatial patterns of the basis CAPs are distinct to each other and reproducible using the proposed shuffled split-half analysis. (A)** Spatial patterns of the basis CAPs in each split-half data. The 4-CAP basis set and the 5-CAP basis set were generated independently from the same split-half data, using the hierarchical clustering across 1,000 shuffled split-half resampling, as described in **S2 Fig**. **(B)** Spatial similarity ($r$, correlation coefficient) of the 4-CAP basis set within the split 1 data (left) and within the split 2 data (right). $r$ values were rounded to the nearest 2 decimal digits for visualization. **(C)** Spatial similarity of the 5-CAP basis set within the split 1 data (left) and within the split 2 data (right). **(D)** Spatial similarity of the 4-CAP basis set between the split 1 and 2 data (left) and of the 5-CAP basis set between the split 1 and 2 data (right). **(E)** Spatial similarity between the 4-CAP basis set and the 5-CAP basis set within the split 1 data (left) and within the split 2 data (right).
(TIF)

**S6 Fig. Spatial patterns of the CAPs estimated from both splits are reproducible and strongly correlated with at least one of the basis CAPs. (A)** From left to right, the marginal distributions of $r$ between all estimated CAPs (ECs) and each basis CAP (BC) from the 4-CAP basis set are illustrated using kernel density estimation. Results were obtained from the split 1 data (top) and the split 2 data (bottom). Each $r$ value is color-coded using a sorting algorithm to label the corresponding EC using the maximum spatial correlation with BCs. **(B)** From left to right, the marginal distributions of $r$ between all estimated CAPs and each BC from the 5-CAP basis set are illustrated using kernel density estimation.
(TIF)

**S7 Fig. The spatial topography of CAP state III is reproducible when it is found in one split and not in another across permutations.**
(TIF)

**S8 Fig. The distribution of correlations between individual fMRI time-frames and the estimated basis CAPs (cluster centroid), to which individual time-frames were assigned by K-means clustering.**
(TIF)

**S9 Fig. Stability of individual mean DT, var DT and FO across permutations.**
(TIF)

**S10 Fig. Between-day reliability of neural measures at single-subject level.** Each datapoint in the scatter plot is a subject. For each subject, neural measures were averaged across permutations.
(TIF)

**S11 Fig. Similarity of temporal organizations between positive and negative co-activation patterns.** CAP states I and II have similar FO, mean DT and DT variance across the positive

and negative co-activation states (I+ vs. I- and II+ vs. II-). Each data point indicate a subject. The temporal metric values across all permutations and 2 days were averaged within each subject.
(TIF)

**S12 Fig. Within-subject variance of FO across 5 CAPs across permutations.**
(TIF)

**S13 Fig. Distribution of individual neural measures of spatiotemporal CAP dynamics differ between subgroups.** The distributions of individual FO, mean DT, and var DT of each CAP state are color-coded by the 3 subgroups. Results from days 1 and 2 data are shown separately and compared between groups. Each data point indicates a subject. Blue lines: $p$-values with Bonferroni-correction across 5 CAPs are estimated using two-sided two-sample $t$ tests between groups, $p_{BON}<.001$ (bold) and $p_{BON}<.05$ (dotted).
(TIF)

**S14 Fig. List of behavioral variables.** Behavioral variable names are identical to the variable names provided by the HCP data dictionary for the S1200 data release: HCP_S1200_DataDictionary_April_20_2018.csv. Check https://wiki.humanconnectome.org/display/PublicData/HCP-YA+Data+Dictionary-+Updated+for+the+1200+Subject+Release for details.
(TIF)

**S15 Fig. Null data were generated by shuffling individual subjects in behavioral data.** Null distributions of partial $R^2$ were estimated for each predictor in the neuro-behavioral association model trained from a split data across 1,000 split-half permutations.
(TIF)

**S16 Fig. Split-half permutation based cross-validation of the prediction model of predicting behavioral PC 1 from neural PCs.** The multiple linear regression models were trained using split 1 data and tested on split 2 data in each permutation. Null data were generated by shuffling individual subjects in behavioral data. The data used to generate the results can be found in **S3 Data**.
(TIF)

**S17 Fig. Repeating the analysis of neuro-behavioral association excluding subjects with high motion did not change the results.** Among 337 subjects, 28 subjects with excessive motion ($FD > 0.5$mm) were excluded. Across 1,000 permutations, a split of subjects ($n = 154$) was randomly selected, and PCA was performed on neural measures from these subjects. Multiple linear regression models for predicting behavioral PCs from these subjects were estimated. Null data were generated by shuffling individual subjects in behavioral data. The data used to generate the results can be found in **S5 Data**.
(TIF)

**S1 Data. Data used to generate Fig 3C.** Three data sets for fractional occupancy, within-subject mean and standard deviation of dwell time are provided. Column A: CAP identification (0: CAP I+, 1:CAP I-, 2: CAP II+, 3: CAP II-, 4:CAP III). Column B: Split 1 or 2. Column C: Permutation index (1–1,000), Column D: Intraclass Correlation Coefficient (ICC).
(XLS)

**S2 Data. Data used to generate Figs 5 and 6.** Sheet 1 (Behavioral PC Loadings) involves the loadings of 262 behavioral variables (columns) on the behavioral PCs 1–15 (rows). Sheet 2 (Individual PC scores) involves the scores of individual subjects (rows) for the behavioral PCs

1–15 (columns).
(XLS)

**S3 Data. Data used to generate Figs 6, S15 and S16.** (Sheets 1–3) Results in **Figs 6** and **S16**. (Sheet 4–6) Null data results in **S15** and **S16** Figs. Columns A-G involve results from the first analysis using split-half datasets across 1,000 permutations. Column A (permutation index). Columns B-F: Partial $R^2$ estimated for neural PC 1, neural PC 2, neural PC 3, age and sex. Column G: Overall $R^2$. Columns H-J involve results from the second analysis using cross-validation, where the multiple linear regression model from Split 1 data was used for predicting individual behavioral PC scores in the corresponding split 2 data across permutations. Column H: Correlation coefficient. Column I: $p$-value estimated for the correlation coefficient. Column J: Overall $R^2$.
(XLS)

**S4 Data. Data used to generate Fig 7.** Column A (subgroup-30nf): Subgroup identified for each individual using 30 neural features. Column B (subject ID). Column C (k5-k4 counts-mean12-z): Probability of CAP III occurrence (z). Column D-F. Individual scores on neural PCs 1–3. Column G. Individual scores on behavioral PC 1.
(XLS)

**S5 Data. Data used to generate S17 Fig.** (Sheets 1–3) Results excluding subjects with high motion. (Sheet 4–6) Null data results excluding subjects with high motion.
(XLS)

## Acknowledgments

We thank Zailyn Tamayo and Mara Heneks for technical supports on the use of computational resources at the division of Neurocognition, Neurocomputation and Neurogenetics (N3) in the department of psychiatry, Yale University School of Medicine.

## Author Contributions

**Conceptualization:** Kangjoo Lee, Alan Anticevic.

**Data curation:** Jie Lisa Ji.

**Formal analysis:** Kangjoo Lee, Jie Lisa Ji.

**Funding acquisition:** John H. Krystal, John D. Murray, Alan Anticevic.

**Investigation:** Kangjoo Lee, John H. Krystal, John D. Murray, Alan Anticevic.

**Methodology:** Kangjoo Lee, Jie Lisa Ji, Grega Repovš, John H. Krystal, John D. Murray, Alan Anticevic.

**Resources:** Jie Lisa Ji, Lining Pan, John D. Murray, Alan Anticevic.

**Software:** Kangjoo Lee, Lining Pan.

**Supervision:** John D. Murray, Alan Anticevic.

**Validation:** Kangjoo Lee, Alan Anticevic.

**Visualization:** Kangjoo Lee.

**Writing – original draft:** Kangjoo Lee.

**Writing – review & editing:** Kangjoo Lee, Jie Lisa Ji, Clara Fonteneau, Lucie Berkovitch, Masih Rahmati, Grega Repovš, John H. Krystal, John D. Murray, Alan Anticevic.

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
