## [Editor Report · Decision Letter 0]

17 Oct 2023

Dear Kangjoo,

Thank you for submitting your manuscript entitled "Human brain state dynamics reflect individual neuro-phenotypes" for consideration as a Research Article by PLOS Biology.

Your manuscript has now been evaluated by the PLOS Biology editorial staff and I am writing to let you know that we would like to send your submission out for external peer review.

Once your full submission is complete, your paper will undergo a series of checks in preparation for peer review. After your manuscript has passed the checks it will be sent out for review. To provide the metadata for your submission, please Login to Editorial Manager (https://www.editorialmanager.com/pbiology) within two working days, i.e. by Oct 19 2023 11:59PM.

Kind regards,

Christian

Christian Schnell, PhD

Senior Editor

PLOS Biology

cschnell@plos.org

---

## [Decision Letter · Decision Letter 1]

15 Dec 2023

Dear Kangjoo,

Thank you for your patience while your manuscript "Human brain state dynamics reflect individual neuro-phenotypes" was peer-reviewed at PLOS Biology. It has now been evaluated by the PLOS Biology editors, an Academic Editor with relevant expertise, and by several independent reviewers. 

In light of the reviews, which you will find at the end of this email, we would like to invite you to revise the work to thoroughly address the reviewers' reports.

As you will see below, the reviewers agree that the topic of the study is interesting, that the study is very well executed and provides important insights. However, there are a few of methodological and technical concerns that need to be addressed. Given the extent of revision needed, we cannot make a decision about publication until we have seen the revised manuscript and your response to the reviewers' comments. Your revised manuscript is likely to be sent for further evaluation by all or a subset of the reviewers.

**IMPORTANT - SUBMITTING YOUR REVISION**

*Re-submission Checklist*

*Published Peer Review*

*PLOS Data Policy*

*Blot and Gel Data Policy*

Sincerely,

Christian

Christian Schnell, PhD

Senior Editor

PLOS Biology

cschnell@plos.org

REVIEWS:

Reviewer #1: Co-activation patterns of BOLD signal have been shown to reflect state-like fluctuations in individuals, and linked to disorder using group-level models. In the current work Lee et al studied CAPs in the human connectome project young adult sample and extend the CAP framework to an individualised approach. They found 4 to 5 unique caps that varied across individuals in a meaningful way, with moderate reliability across sessions. Principle component in CAP variation could be linked to a principle component of behaviour variation in the sample. 

Overall I think the work is interesting, and I have a few questions/clarifications that I have summarised below. I hope they make sense.

1. Were the subjects unrelated? If yes/no are CAPs more similar between MZ vs DZ twins? 

2. Is the visualisation in 3B correct? I.e. very similar profiles between day 1 split 1 and 2 and day 2 split 1 and 2 but not between?

3. Overall the manuscript moves towards more general patterns at the end of the result. While it is interesting, I did wonder a little what then these reduced dimensions represent and whether these CAP / behavioral features are particular for HCP? 

4. The authors include the full BOLD signal range, did they also test for the impact of this decision by focussing on different frequency ranges? 

5. In the introduction the CAP framework as framed as a compromise between state and trait-like patterns in the brain, but in the final figures CAPs are further put into a trait-like perspective, also linking to behaviours. How to interpret this?

6. The concept of reproducibility seems core to the hypothesis of the work, yet ICC for individuals is fairly low, what would this mean for the framework? Does this also relate to the state-trait conundrum? 

7. moreover another aim is to link CAP to behavioral phenotypes across multiple behavioral domains, yet in the end only the principle component is selected? 

8. To what extend does the spatial alignment of co-activation patterns in individuals contributes to the observed cap variability? 

Reviewer #2: The manuscript by Lee and colleagues characterized recurring co-activation patterns in resting-state fMRI activity and related their features with cognitive/behavioral phenotypes of participants in the human connectome project dataset. The study demonstrates that combining state and trait level variances in neural activity can be a useful neuromarker of inter-individual variance. The study was timely and was rigorously done. The purpose of my comments is to help improve the manuscript - so please feel free to counter-argue.

The authors emphasize the use of CAPs (i.e., low-dimensional brain states) in characterizing state-trait variance. Though the authors review past CAPs literature on patient population, the authors omitted quite many seminal studies that identified latent brain states—including features such as fractional occupancy, dwell times, etc—and related with state and trait variances. I listed some of the work, just to name a few; Vidaurre et al. (2017) even used the same HCP trait measures. I agree that the majority of work (especially initial work) focused on trait-level neural characteristics to relate with individual phenotypes (e.g., Finn et al., 2015); however, using neural state dynamics or time-varying functional connectivity to understand individual differences is also an active area of research. Please discuss the current manuscript in relation to past literature.

Vidaurre, Smith & Woolrich, 2017, PNAS (https://doi.org/10.1073/pnas.1705120114): various trait measures in HCP

Shine et al., 2019, Nat Neurosci (https://doi.org/10.1038/s41593-018-0312-0): fluid intelligence

Taghia et al., 2018, Nat Commun (https://doi.org/10.1038/s41467-018-04723-6): working memory performance

The first sentence of the abstract needs to be reworded: "Neural activity and behavior manifest state and trait dynamics, as well as variation within and between individuals". My first impression was: aren't "state and trait dynamics" and "variation within and between individuals" the same thing? My second thought was: that variation across individuals (e.g., people have different static functional connectivity patterns) and trait dynamics (e.g., people are characterized by some recurring dynamics) could potentially be different, but how is variation within individuals and state dynamics different? Either way of interpretation, this sentence is confusing.

The main finding of this paper is Fig. 6G. Is this result robust? To test for robustness/predictability, could the authors, for example, conduct neural and behavioral PCA on split-half, fit the regression model, and apply these to the other split-half by transforming the values to PC space and testing if the regression output is predictable?

The authors argue that the study "demonstrate a reproducible estimation of spatio-temporal CAP features at the single-subject level". However, there is a nuance - whether the results are truly at the "single-subject level". The basis set of CAPs needed to be estimated on the concatenated group time series. I do not know to what extent CAPs generated at the single-subject level are comparable to the basis set of CAPs. The features (e.g., FO, mean DT, var DT, across-day reliability) may be generated at individual subject level, but because the model was built from the aggregated group data from the start, I am unsure to what extent this sentence is warranted.

[Comments on figures]

Overall, the figures contain too much information. There are many parts that are unnecessary in delivering the main findings. I suggest removing many figures that assess the validity of the models or which purely describe observations that should be observed by the nature of the model. For example, Fig. 5 could be moved to supplement. The three distributions "should" be different across subgroups because the clustering algorithm found clusters that accentuate differences. Compared to the information it conveys, it's a too-busy figure. Fig. 1B-E could also be removed/moved to supplement because the importance of these figures is much less than Fig. 1G or 1H.

I consider Fig. 7A-C misleading. The probability of CAP 3 occurrence "should" be associated with neural PCs because neural PCs were derived from CAP features which include the FO of CAP 3. Thus, significant relationships between CAP 3 occurrence and neural PCs are expected. However, with these figures visualized as such in the main figure, it may give a misleading impression that this is an interesting finding rather than the description of the neural PCs. On the other hand, Fig 7D is a finding. However, the sentence, "These results together indicate that the spatio-temporal properties of CAP 3 contribute to the positive correlation between the neural PC 1 and the behavioral PC 1 (Fig. 6G).", is not warranted because the contribution of CAP 3 FO in neural PC1 - behavioral PC 1 relationship was not directly tested by the statistical model.

In Fig. 2, shouldn't Fig 2A have 4 rows (ECi, with i = 1 to 4) and 5 columns instead of 5 x 5? Also, why is it 502 permutations?

[Comments on methods]

The authors succinctly described their analyses in the methods and used Figs. S1 and S2 to detail their workflow. However, it did not contain much of a "narrative". I had a hard time understanding why such a method was selected, why such a step was important, how such a method was used to test what, etc. I think the manuscript would improve in terms of readability if the methods were explained rather than simply described.

When estimating CAPs, the authors used a shuffled split-half resampling strategy across multiple permutations. However, what was the motivation for taking this approach, rather than, for example, applying K-means clustering just once to the concatenated time series of the entire sample? For the permutation, is the reason because K-means clustering estimation gave different outputs every time it was run? Is split-half analysis due to test for reliability? Please justify in the manuscript.

In Methods, "Lastly, an k-CAP basis set was obtained by using the agglomerative hierarchical clustering of the CAPs estimated from all permutations (Supplementary Fig. S2)." Following this sentence, please describe the steps conceptually. This is an important method but Fig. S2 is insufficient to deliver intuitively what has been done.

Explain the elbow method that was used to determine the optimal K. What criteria determined the "elbow"?

[Minor comments]

Figs 2 and 3 highlighted the reliability of CAP states and their features. Additionally, I was curious about the FO, mean DT, and var DT distributions of three CAP states. For example, were the fractional occupancies of CAPs hugely varied across participants? Were the fractional occupancies of CAPs 1 and 2 much higher than the fractional occupancy of CAP 3 in general?

Pg 5. Left column "The correlation of individual neural measures between day 1 and day 2 was …": Are these r values of between-days reliability significant, when compared to a null distribution (non-parametric permutation test)? Also, how are these values (mean r = 0.41, 0.41, and 0.38 for FO, mean DT, and var DT) different from the values reported in Fig. 3D (r = 0.5, 0.45, and 0.41)?

Fig. 3B: The figure well illustrates across-permutation reliability and across-split reliability. However, if I'm understanding correctly, if between-day reliability is high - as roughly r = 0.4 - then shouldn't the Day 2 color scale roughly show a green-red-blue gradient from top to bottom? (or is it already showing but I'm just not seeing the gradation?) 

Pg 3. Last sentence of the left column: change "cosign similarity" to "cosine similarity".

For simplicity, I don't find the need to additionally define "time-consecutive segment (c)". Dwell times can be explained without the "additional" explanation of the time-consecutive segment.

Reviewer #3: In "Human brain state dynamics reflect individual neuro-phenotypes" the authors use a novel clustering approach to delineate state-trait variations in neural dynamics and define links inter-individual variability in behavior and cognition. The work is methodologically rigorous and quite thorough. The findings reported are an important contribution to the literature and beautifully displayed with appropriate visualizations. We have relatively minor comments for the authors to consider; we have signed our names for transparency.

-Golia Shafiei & Ted Satterthwaite

(1) Methods detail. We were surprised that the methods were fairly sparse. While many of the methods were introduced in the results, they received relatively dedicated treatment in the methods section. This section could be expanded (either in the main text or the supplement) to allow readers to understand the approach in greater detail. For example, there is limited detail regarding the clustering algorithms and the associated parameters. (E.g., what was the distance metric used for agglomerative clustering?). 

(2) Underlying biology. Although the analytical approach is a tour de force and is an important contribution to the field by itself, links to underlying biology received relatively scant attention. Contextualizing results in the context of systems/developmental neuroscience could further increase the impact of this study. 

(3) Clustering quality. It is somewhat unclear what the quality of the clustering solution was. Did the authors consider quantifying this using a measure such as silhouette score? The same comment applies to the K-means solution for the CAP analysis.

(4) Temporal censoring in preprocessing. The analysis workflow depicted in Fig S1 indicates that scrubbing was applied for the time frame selection procedure; this is not mentioned in the methods section. How did the scrubbing process influence the findings of CAPs analysis, especially analyses of state transitions & dwell time, which would seem likely to be heavily impacted by the timeseries discontinuities introduced by scrubbing?

(5) Individual differences in motion. Relatedly, did residual motion after scrubbing relate to the CAPS themselves, the subgroups, or the brain-behavior relationships? Including mean FD as a covariate would enhance confidence. 

(6) Subcortical regions. It seems like subcortical regions were also included in the initial CAPs analysis, given that the Cole-Anticevic Brain Network Parcellation includes cortical surface and subcortical volume parcels. However, it is not clear whether the subcortical regions were included in the subsequent analysis. Could the authors comment on how subcortical data was handled and why it was excluded from the results?

Reviewer #4: Thank you for the opportunity to review "Human brain state dynamics reflect individual neuro-phenotypes" by Lee and colleagues. The manuscript describes an extensive investigation of patterns of recurrent topographies in spontaneous neural activity measured by resting state fMRI. The authors apply the well-studied CAP method to derive 4/5 canonical patterns and show that they are mostly reproducible in individual participants and that they are correlated with individual differences in behavior.

Although the premise is interesting, I was not enthusiastic about the method, or whether the findings constitute a conceptual advance. As I outline below, I find it hard to believe that 15-minute chunks of neural recordings only contain information about 4/5 underlying processes. The patterns themselves do not appear to be highly differentiated from each other, or from the dominant PC1 in resting state FC. 

- Why make the assumption that only one CAP is present in any single volume? Surely multiple sensory, cognitive and motor processes can take place simultaneously.

- On a related note, the authors find that there are an average of 4-5 CAPs per participant. This, like other discrete-state methods used in neuroscience, strikes me as counter-intuitive, in the sense that are probably many many more simultaneous processes taking place during an average 15-minute scan.

- Many of the CAPs are highly anatomically similar, and also generally similar to the frequently-studied principal gradient of FC. There is quite a bit of attention on resting state networks but clearly the FC PC1 is the dominant signature in all these patterns.

- More generally, it is unclear what these patterns represent biologically. Are they related to the underlying micro- or macro-scale anatomy? 

- The motivation presented in the Introduction suggests that CAPs could better capture individual differences in behavior because they are sensitive to both state and trait. Yet there is no clear comparison between the two.

---

## [Decision Letter · Decision Letter 2]

19 Jul 2024

Dear Kangjoo,

Thank you for your patience while we considered your revised manuscript "Human brain state dynamics reflect individual neuro-phenotypes" for publication as a Research Article at PLOS Biology. This revised version of your manuscript has been evaluated by the PLOS Biology editors, the Academic Editor and the original reviewers.

Based on the reviews, we are likely to accept this manuscript for publication, provided you satisfactorily address the remaining comments from the reviewers and the following data and other policy-related requests:

* We would like to suggest a different title to improve its accessibility: "Human brain state dynamics are highly reproducible and associated with neural and behavioral features"

* Please add the links to the funding agencies in the Financial Disclosure statement in the manuscript details

* All research involving human participants must have been conducted according to the principles expressed in the Declaration of Helsinki. Please state in the manuscript that this is the case for your study (if correct).

* DATA POLICY:

Regardless of the method selected, please ensure that you provide the individual numerical values that underlie the summary data displayed in the following figure panels as they are essential for readers to assess your analysis and to reproduce it: 3C, 5C, 6DEFGH, 7A and similar panels in the supplementary figures.

* CODE POLICY

* Please note that per journal policy, we do not allow the mention of "data not shown", "personal communication", "manuscript in preparation" or other references to data that is not publicly available or contained within this manuscript. Please either remove mention of these data or provide figures presenting the results and the data underlying the figure(s).

We expect to receive your revised manuscript within two weeks. 

*Published Peer Review History*

*Press*

Sincerely,

Christian

Christian Schnell, PhD

Senior Editor

cschnell@plos.org

PLOS Biology

Reviewer #1: I think the authors for their thorough answers!

Reviewer #2 (Hayoung Song): I thank the authors for addressing my earlier comments thoroughly. The manuscript has greatly improved in terms of rigor, details, and narratives. I have no further comment and believe the manuscript is in good standing for publication. 

Reviewer #3 (Ted Satterthwaite & Golia Shafiei): All of our comments have been largely addressed. One final point that could merit clarification (but should not delay publication) is in regards to scrubbing. First, odes not appear to be described in the (appropriately expanded) new methods section. Second, it is somewhat unclear how a scrubbing interval interacts with measures such as dwell time. If a consistent state is interrupted by a high motion volume and scrubbing is applied, is that considered a state transition, or does the dwell time continue? While the authors analyses suggest that it is unlikely that these scrubbing events impact the results, such methodological details would be nice for readers to be able to refer to.

Reviewer #4: The authors have addressed my concerns to the best of their knowledge, and in line with the standards in the field. I am therefore happy to defer to the Editor and other reviewers. However, I still believe that the biological interpretation of these CAPs is ambiguous and that the assumptions imposed (a small number of discrete states) are unrealistic.

---

## [Editor Report · Decision Letter 3]

15 Aug 2024

Dear Kangjoo,

Thank you for the submission of your revised Research Article "Human brain state dynamics are highly reproducible and associated with neural and behavioral features" for publication in PLOS Biology. On behalf of my colleagues and the Academic Editor, Laura Lewis, I am pleased to say that we can in principle accept your manuscript for publication, provided you address any remaining formatting and reporting issues. These will be detailed in an email you should receive within 2-3 business days from our colleagues in the journal operations team; no action is required from you until then. Please note that we will not be able to formally accept your manuscript and schedule it for publication until you have completed any requested changes.

PRESS

Sincerely, 

Christian

Christian Schnell, PhD

Senior Editor

PLOS Biology

cschnell@plos.org